# Peptidoglycan recruitment by a penicillin binding protein

Yamanappa Hunashal [1,6,9], Matthieu Fonvielle [2,7,9],
Masumi Takayama Kobayashi[1], Meng S. Choy[1], Ganesan Senthil Kumar [1,3],
Paul Ugalde Silva [4,8], Yucheng Liang [2], Charlene Desbonnet[4], Louis B. Rice[4],
Michel Arthur [2], Rebecca Page [5] & Wolfgang Peti [1] ✉

The cell wall is essential for bacterial survival. Its core component is peptidoglycan (PG), a polymer comprised of disaccharide-peptides (stem peptides) that are cross-linked to one another via transpeptidation by penicillin-binding proteins (PBPs). While much is known about how PBPs are inactivated by β-lactam antibiotics, little is known about how PBPs bind and catalyze the transpeptidation of PG. Here we show how native PG and stem peptides are recruited to PBP5 of *E. faecium*, a critical ESKAPE pathogen. We discovered that PG binds PBP5 at the periphery of the PBP active site cleft, not the active site, and that the D-Ala leaving group contributes minimally to PBP binding. We show that β-lactam antibiotics and stem peptides can bind PBP5 simultaneously. We also show that only the single central residue of the stem peptide (L-Lys substituted by D-iAsn in *E. faecium*) is both necessary and sufficient for peptide recruitment. Finally, we translate our molecular findings by demonstrating that recruitment binding variants are unable to create a PG cell wall in *E. faecium*. Our studies define the key molecular interactions that govern bacterial cell wall formation and provide opportunities for the development of antibiotics that do not rely on PBP inactivation.

Peptidoglycan (PG) is a crucial component of the bacterial cell wall, a structure that protects bacterial cells against the turgor pressure of the cytoplasm[1,2]. PG is composed of amino sugar chains that are cross-linked to peptides. Synthesis of a PG subunit is initiated by the production of UDP-linked *N*-acetyl-D-glucosamine (NAG). NAG is converted by MurA/B enzymes into *N*-acetylmuramic acid (NAM), which is further modified by MurC/D/E/F enzymes that incorporate proteogenic (L-Ala and L-Lys) and non-proteogenic (D-Glu and D-Ala) amino acids to form a stem peptide (SP; Fig. 1a). Assembly of the disaccharide-peptide subunit is completed on the precursor linked to the

undecaprenyl-lipid carrier by a pyrophosphate bond. These maturation steps involve the addition of NAM, the addition of D-Asp to L-Lys, and amidation of the α-carboxyl of D-iGlu and D-iAsp to form a biosynthetic intermediate commonly referred to as Lipid II, which is flipped from the cytosol to the periplasm and polymerized by transglycosylation and transpeptidation reactions. The latter is catalyzed by a family of enzymes known as penicillin-binding proteins (PBPs), which are the essential targets of β-lactam antibiotics such as penicillin. PG cross-linking by PBP transpeptidases is achieved in two steps[1]. First, the catalytic serine attacks the carbonyl of the penultimate

[1]Department of Molecular Biology and Biophysics, University of Connecticut Health, Farmington, USA. [2]INSERM ERL 1336, UMRS 8228, Sorbonne Université-ENS-PSL-CNRS, Paris, France. [3]National Institute of Immunology, New Delhi, India. [4]Department of Medicine, Rhode Island Hospital, Warren Alpert Medical School of Brown University, Providence, USA. [5]Department of Cell Biology, University of Connecticut Health, Farmington, USA. [6]Present address: Division of Science, New York University Abu Dhabi, Abu Dhabi, United Arab Emirates. [7]Present address: Institute for Integrative Biology of the Cell (I2BC), CEA, CNRS, Université Paris-Saclay, Gif-sur-Yvette, France. [8]Present address: Department of Microbiology and Infectious Diseases, Faculty of Medicine and Health Sciences, Université de Sherbrooke, Sherbrooke, QC, Canada. [9]These authors contributed equally: Yamanappa Hunashal, Matthieu Fonvielle. ✉e-mail: peti@uchc.edu

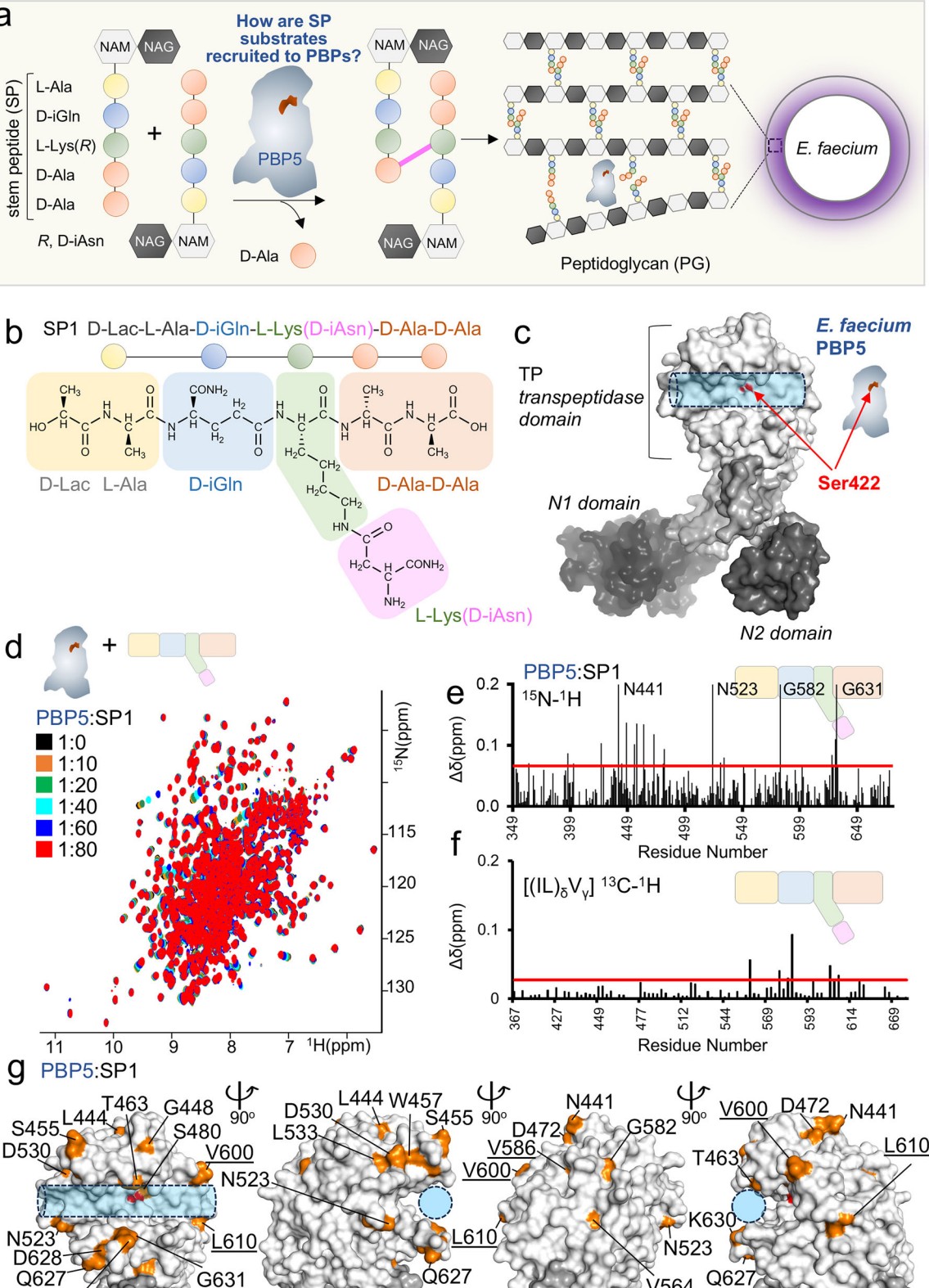

**Fig. 1 | *E. faecium* STEM peptides are recruited by PBP5. a** Role of PBP5 in *E. faecium* peptidoglycan (PG) formation through its transpeptidase activity. **b** *E. faecium* SP (SP1). **c** PBP5 structure; surface representation (PDBid: 6MKA); transpeptidase domain (TP) and catalytic Ser422 (red) are highlighted. **d** Overlay of 2D [¹H,¹⁵N] TROSY spectrum of PBP5 (black) with increasing concentrations of SP1 (up to 1:80 PBP5:SP1 ratio; orange to red). **e** ¹H/¹⁵N chemical shift perturbation (CSP) vs residue number plot for SP1:PBP5 (80:1 molar ratio); average+1σ (red line) for the PBP5 TP domain. **f** ¹H/¹³C ILV methyl CSPs vs residue number plot for SP1:PBP5 (80:1); average+1σ (red line) for the PBP5 TP domain. **g** SP1 CSPs mapped on PBP5 structure (gray surface; PDBid: 6MKA); orange surface: residues with significant changes (underlined residues indicate ¹H/¹³C ILV methyl data). S422 is highlighted in red. Transparent blue cylinder highlights active site groove.

D-Ala residue (4th position of the SP) of an acyl donor stem peptide. This releases the C-terminal D-Ala and results in a covalent acyl-enzyme adduct with the donor peptide. In a second step, the carbonyl of the D-Ala adduct undergoes nucleophilic attack from a primary amine located at the extremity of a residue side chain (3rd position of the SP) of an acceptor stem peptide (a second molecule of SP)[3]. This generates a covalent cross-link between the two SPs, linking the glycan strands to one another. While much is known about how PBPs are inactivated by β-lactam antibiotics[4], a molecular understanding of how PBPs recognize and catalyze the transpeptidation of their endogenous substrates remains an open question.

β-lactam antibiotics acylate and inactivate the catalytic serine of PBPs[3]. Because β-lactams are proposed to function as structural analogs of the D-Ala-D-Ala moieties of stem peptide substrates, it has been reasoned that the same PBP residues mediate SP and β-lactam recruitment[5]. Additionally, while PG/SP composition is generally conserved within bacteria from the same species, it is divergent between species[1,6]. The most significant differences in SP composition occur in the 3rd and 5th positions. Position 3 is often a modified L-Lys (modifications can be significantly different, including substitutions by D-iAsn or a pentaglycine side chain) or a modified diaminopimelic acid (DAP). Position 5 is D-Ala, D-Ser, or D-Lac. If and how PBPs from different species distinguish amongst distinct SPs is unknown[7].

Here we study the low-affinity class B PBP from *Enterococcus faecium*, an ESKAPE pathogen that is a leading cause of nosocomial infections[8]. This PBP, hereafter referred to as PBP5, is responsible for the intrinsic resistance to β-lactams in *E. faecium*[9,10], conveying resistance to high levels of cephalosporins and moderate levels of penicillins. Under the selective pressure of β-lactams, *E. faecium* clinical isolates acquire resistance to penicillins by modification of PBP5, thereby abolishing the therapeutic efficacy of these drugs[10]. Thus, defining the molecular basis of the recognition of the cross-linking of PG/SP is essential for developing antibiotics that exhibit high barriers to resistance.

Here, we used biochemistry, microbiology, structural and chemical biology to define how *E. faecium* PG is recruited to PBP5. We discovered that the PBP5 residues that mediate PG/SP recruitment, a step that precedes cross-linking, are distal from the nucleophilic Ser422 and distinct from most mutations that are responsible for increased β-lactam resistance. Thus, the PBP5 variants resistant to β-lactams are fully capable of PG recruitment, explaining in part why PG cross-linking is not negatively impacted by mutations responsible for the acquisition of β-lactam resistance. We confirmed these molecular results with physiological *E. faecium* assays. We then used complimentary structural experiments to identify which residues of the stem peptide substrates bind PBP5 during substrate recruitment. Unexpectedly, we found that only a few stem peptide atoms engage PBP5, with only the central stem peptide modified lysine residue (L-Lys(D-iAsn)) being most important for binding. We then confirmed these data using a set of systematically chemically modified SP peptides. Finally, we showed that experimentally derived PBP5 recruitment binding variants are unable to create a PG cell wall. Together, these studies reveal why SPs are not specific to a particular PBP[7], why β-lactam resistant variants[10] still effectively process SP cross-linking reactions and provide a innovative approach for the development of antibiotics with high barriers to resistance by blocking SP substrate recruitment.

## Results

### *E. faecium* stem peptides are recruited to PBP5 by residues at the periphery of the active site

We previously described the optimization of *E. faecium* PBP5 (678 aa; 73.6 kDa; PBP5$_{37-678}$ soluble form hereafter referred to as PBP5; $T_m$ = 333 K) for solution-state NMR spectroscopy experiments[11]. We also established procedures to extract PG from *E. faecium* in yields that allow for molecular analysis, and developed sophisticated synthesis and purification methods to generate bacterial stem peptides (SPs) of defined composition[12]. Here, we combined these advances to molecularly define how PBP5 specifically and PBPs generally bind and cross-link their respective SP substrates, activities that are strictly required for an osmoprotective bacterial cell wall.

PBP5 catalyzes the cross-linking of two SPs (a donor and an acceptor). To define the molecular interactions between the *E. faecium* SP (the donor and acceptor SP1 molecules, D-Lac-L-Ala-D-iGln-L-Lys(D-iAsn)-D-Ala-D-Ala; Fig. 1b) and PBP5 (Fig. 1c), we used 2D [$^1$H,$^{15}$N] transverse relaxation optimized spectroscopy (TROSY) and 2D [$^1$H,$^{13}$C] ILV methyl heteronuclear multiple quantum coherence (HMQC) measurements with increasing ratios SP1 vs PBP5 (Fig. 1d, Supplementary Fig. 1). Chemical shift perturbations (CSPs) in PBP5 residues were readily identified, all of which were in a fast exchange regime (Fig. 1e, f). For practical reasons (synthesis of peptides), most titrations were stopped at a ratio of 1:80 (PBP5:SP peptide), with most CSPs still not in full saturation, highlighting a weak interaction ($K_D$ ~ 3.9 ± 1 mM, Supplementary Table 1). The weak affinity is likely compensated for in vivo by: (a) high local substrate concentrations at the outer surface of the cytoplasmic membrane, (b) precursor channeling in multi-enzyme complexes, (c) contributions of the modified sugar moieties or (d) combinations of the above. Significant CSPs were identified for multiple residues, especially in the transpeptidase (TP) domain, that cluster into distinct regions (Fig. 1g). First, we observed CSPs for residues Thr463 and Ser480. These residues are at the PBP5 active site and thus likely contribute to active site recruitment. The second region is the back side of the TP domain (Leu444, Gly448, Asn441, Asp472). The CSPs observed for these residues likely reflect conformational changes at the PBP5 active site during recruitment. The third, most extensive region, is defined by the outer edges of the active site cleft, including PBP5 residues Asn523, Leu533, Asp530, Trp457 and Gln627-Gly631 (a flexible loop). Lastly, we also identified CSPs in more distal residues (Glu284, Glu291, Ser294, Asn295, Gln396, Glu397) and in the PBP5 N1 domain (the N1 domain is one of 4 domains in PBPs and links the membrane-inserting helix with the TP domain). We previously observed similar CSPs in the presence of the β-lactam antibiotics penicillin G (penG) and ceftaroline[11] (we showed that these CSPs are mediated by rigid surface loops in the PBP5 TP domain, L1 and L2, that facilitate communication between the TP and the N1 domain; however, they do not influence binding or activity at the PBP5 active site[11]). Because the vast majority of PBP5 resistance mutations are close to the PBP active site[10], our data show that β-lactam antibiotics and SPs bind distinct surfaces of PBP5, explaining, in part, why resistant variants still effectively process peptide cross-linking.

### PBP5 stem peptide recruitment residues also bind endogenous *E. faecium* peptidoglycan

To test if our observations using a model SP also occur with native *E. faecium* substrates, we tested the interaction of PBP5 with peptidoglycan (PG; reduced and oxidized) extracted and purified from *E. faecium* (Fig. 2a). We used 2D [$^1$H,$^{15}$N] TROSY NMR measurements with increasing PBP5 to PG ratios (1:0.25−1:5 ratio) to measure these interactions (Fig. 2b, c). The data showed that oxidized-PG binds more weakly (small CSPs, Fig. 2b, d) than reduced-PG (large CSPs; Fig. 2c, e). Indeed, the reduced-PG NMR data showed CSPs already at sub-stoichiometric ratios and many PBP5 NH$^N$ cross peaks were broadened beyond detection (i.e., missing in the spectrum; intermediate exchange timescale), behavior commonly observed for tighter interactions (Supplementary Fig. 2). While limited, the CSP pattern of oxidized-PG results corresponded to a subset of residues that were also identified in the SP1 interaction (Figs. 1g and 2f). This correspondence was more apparent with reduced-PG, which showed a significantly more expansive CSP interaction pattern, including not only all residues with CSPs identified in the SP1 and oxidized-PG interactions, but many additional residues surrounding the PBP5 active site cleft (Figs. 1g and 2g). Together, these data show that PG exhibits

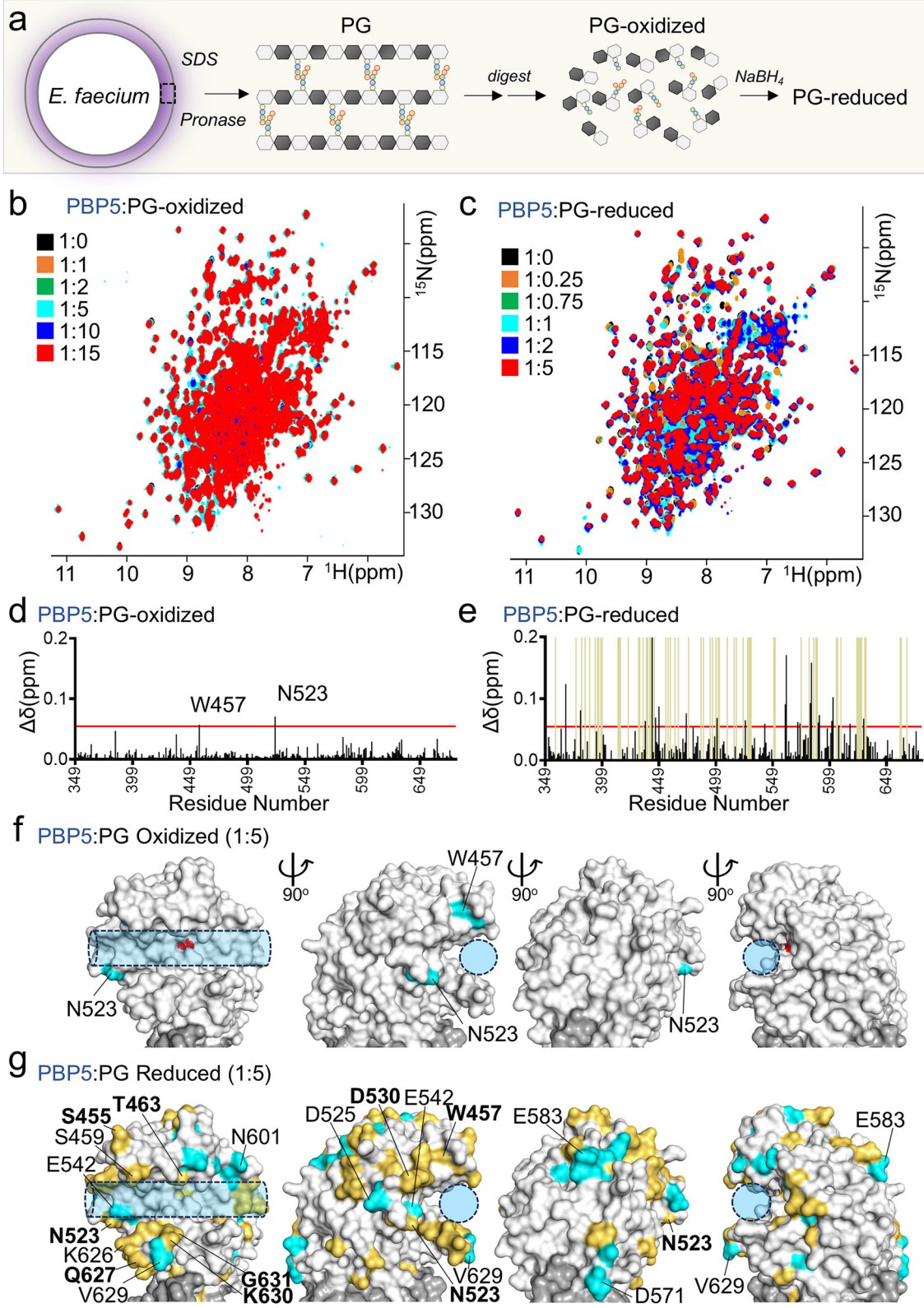

**Fig. 2 | PBP5 recruits *E. faecium* peptidoglycan. a** Peptidoglycan extraction for NMR studies from *E. faecium*. **b** Overlay of 2D [¹H,¹⁵N] TROSY spectrum of PBP5 (black) with increasing concentrations of oxidized PG (up to 1:15 PBP5:PG ratio; orange to red). **c** Overlay of 2D [¹H,¹⁵N] TROSY spectrum of PBP5 (black) with increasing concentrations of reduced PG (up to 1:5 PBP5:PG ratio; orange to red). **d** ¹H/¹⁵N chemical shift perturbation (CSP) vs residue number plot for oxidized PG:PBP5 (5:1 molar ratio); average+1σ (red line) for the PBP5 TP domain. **e** ¹H/¹⁵N

chemical shift perturbation (CSP, black bars) vs residue number plot for reduced PG:PBP5 (5:1 molar ratio); average+1σ (red line) for the PBP5 TP domain. Olive green bars: Disappeared residues. **f** Oxidized PG CSPs mapped on PBP5 structure (gray surface; PDBid: 6MKA); Cyan surface: residues with significant changes. S422 is highlighted in red. **g** Reduced PG CSPs mapped on PBP5 structure. Cyan surface: residues with significant changes. Yellow surface: residues that disappear because of broadening.

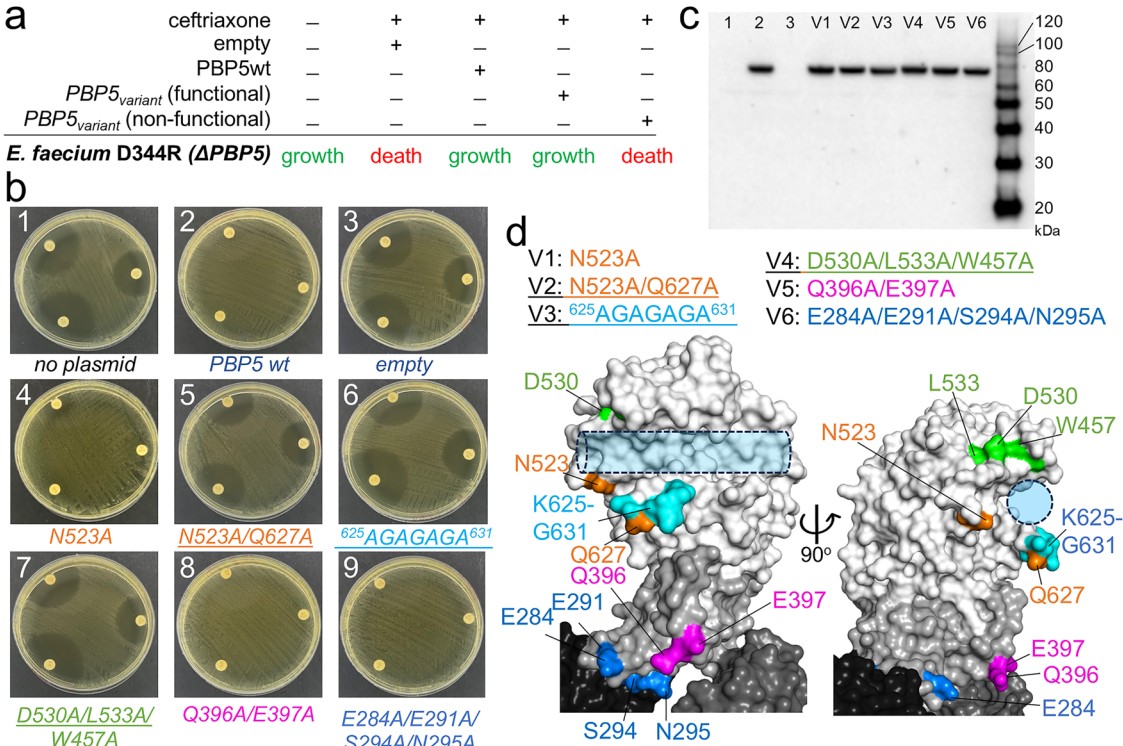

**Fig. 3 | Disruption of PG formation in *E. faecium*. a** Expected result of the disk assay using *E. faecium* strain without PBP5 (D344SRF, ΔPBP5). **b** Disk diffusion assay. 1, *E. faecium* D344SRF vs. ceftriaxone; 2, *E. faecium* D344SRF + PBP5 plasmid vs. ceftriaxone; 3, *E. faecium* D344SRF + empty plasmid vs. ceftriaxone; 4, *E. faecium* D344SRF + PBP5_V1 plasmid vs. ceftriaxone; 5, *E. faecium* D344SRF + PBP5_V2 plasmid vs. ceftriaxone; 6, *E. faecium* D344SRF + PBP5_V3 plasmid vs. ceftriaxone; 7, *E.* *faecium* D344SRF + PBP5_V4 plasmid vs. ceftriaxone; 8, *E. faecium* D344SRF + PBP5_V5 plasmid vs. ceftriaxone; 9, *E. faecium* D344SRF + PBP5_V6 plasmid vs. ceftriaxone. **c** Western Blot for PBP5 protein expression. Lane numbers correspond to experiments in (**b**).: blot was performed *n* = 1. **d** PBP5 variants used in disk assay mapped on PBP5 surface (gray surface; PDBid: 6MKA). Orange: V1 and V2, Cyan: V3, Green: V4, Magenta: V5 and Blue: V6.

enhanced binding versus isolated SP peptides, and, more importantly, that the recruitment sites are identical.

## PG recruitment is essential for *E. faecium* viability

While PBP5 is not essential for *E. faecium* growth, it is essential for the resistance of *E. faecium* to β-lactam antibiotics[9]. This characteristic of PBP5 led to the development of a disk diffusion susceptibility assay in which *E. faecium* growth depends on the expression of functional PBP5 (Fig. 3a). Briefly, all *E. faecium* PBPs except PBP5 are potently inhibited by the β-lactam antibiotic ceftriaxone (PBP5 does not bind ceftriaxone and thus is not inhibited by it, Supplementary Fig. 3a). Thus, a strain lacking the *pbp5* gene (*E. faecium* D344SRF, Fig. 3a–c; Supplementary Fig. 3b) fails to grow in the presence of ceftriaxone, while *E. faecium* D344SRF transformed with a plasmid that expresses PBP5 is not sensitive to ceftriaxone enabling growth in the presence of the drug (Fig. 3a–c; Supplementary Fig. 3b; PBP5-mediated resistance in *E. faecium* is enhanced when the gene encoding PBP5 is present downstream of genes *ftsW* and *psr*, both genes were incorporated into the plasmid upstream of *pbp5*[10]).

Leveraging this assay, we tested the functional relevance of PBP5 recruitment sites identified by NMR. *E. faecium* D344SRF was transformed with plasmids encoding either no PBP5, PBP5 wt or one of six PBP5 variants (V1: N523A; V2: N523A_Q627A; V3: loop 625-631; V4: D530A_L533A_W457A; V5: Q396A_E397A; V6: E284A_E291A_-S294A_N295A; Fig. 3a–d; Supplementary Fig. 3b; expression of plasmid-encoded PBP5 variants in *E. faecium* D344SRF was confirmed by Western blot, Fig. 3c, Supplementary 3c). The data show PBP5 variants V1, V5 and V6 did not impact PBP5 function, i.e., the PG cell wall was formed and *E. faecium* was viable (thus the observed NMR CSPs of PBP5 residues Glu284, Glu291, Ser294, Asn295, Gln396 and Glu397 were due to indirect conformational or dynamic effects and do not contribute to PG binding and cell wall formation). In contrast, PBP5

variants V2, V3 and V4 behaved like the empty vector control, i.e., no PG cell wall was formed and *E. faecium* was not viable (Fig. 3b). Correspondingly, the *E. faecium* D344SRF V2, V3 and V4 variants exhibited susceptibility to all tested β-lactams (Supplementary Table 2) and broth-dilution MIC determinations showed a low MIC of ceftriaxone, even lower than that observed in *E. faecium* D344SRF lacking PBP5 (Supplementary Table 3). Together, these results strongly suggest that PBP5 residues Asn523, Asp530, Leu533, Trp457 and the residues of loop 625-631 are crucial for *E. faecium* PG/SP recruitment, and, further, that this is an obligatory step that precedes transpeptidation.

## D-Ala-D-Ala is dispensable for SP recruitment

We next asked which elements of the *E. faecium* SP interact with PBP5 by performing an inverse interaction experiment using 1D ¹H saturation transfer difference (STD) NMR (Fig. 4a) with SP1 and SP1 derivatives (SP2-6; Fig. 4b). This experiment does not require protein labeling and reliably captures weak interactions (Supplementary Fig. 4a, b)[13,14]. Performing these experiments with SP1 showed that mainly the atoms in D-iGln and L-Lys CH/CH₂ groups contributed significantly to PBP5 binding (STD ≥ 50%; Fig. 4c, Supplementary 4a,b). By comparison, no interactions were observed for atoms in D-Lac or L-Ala and only a single STD was observed for an atom in D-Ala-D-Ala (because all CH atoms of the D-iAsn group of the L-Lys(D-iAsn) overlapped, it was impossible to determine if any D-iAsn atoms interacted with PBP5; Supplementary Fig. 4b).

The lack of strong interactions for D-Ala-D-Ala was unexpected. This is because the 4-membered ring of β-lactam antibiotics are thought to function as substrate mimetics due to the structural similarity between the drug and the peptidyl-D-Ala-D-Ala extremity of the peptidoglycan precursors[5]. To test if the D-Ala-D-Ala component of SP1 is dispensable for PBP5 recruitment, we generated a peptide that lacks the D-Ala-D-Ala moiety (D-Lac-L-Ala-D-iGln-L-Lys(D-iAsn); SP2, Fig. 4b)

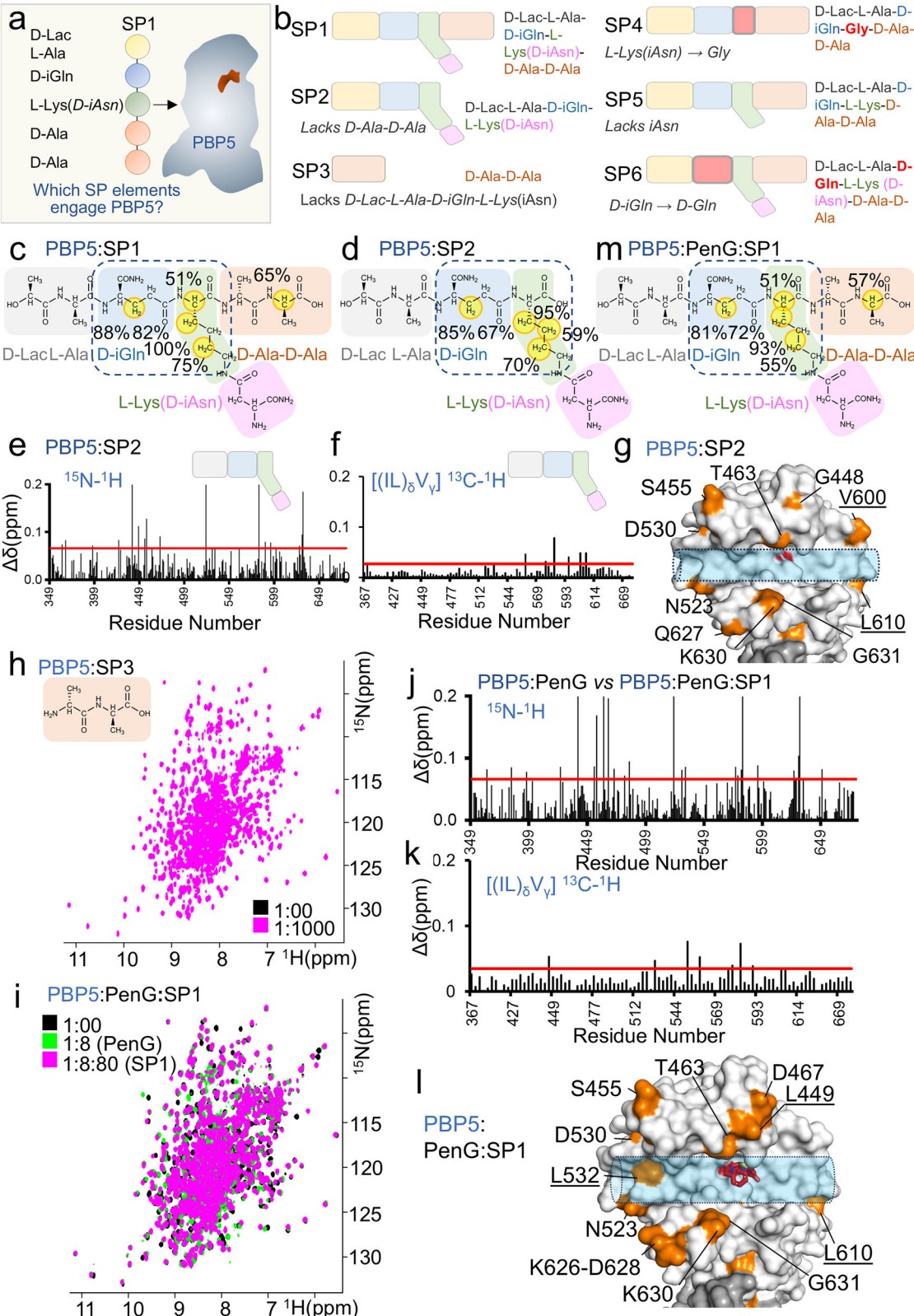

and repeated the CSP and 1D ¹H STD interaction studies. The SP2 1D ¹H STD NMR analysis was nearly identical to that with SP1 (Fig. 4d and Supplementary 5a,b). Consistent with this, the CSP interaction profile of SP2 (lacks D-Ala-D-Ala) with PBP5 was similar to that observed with SP1 (with D-Ala-D-Ala; Fig. 4e−g and Supplementary 5c,d). Both sets of results demonstrate that the D-Ala-D-Ala moiety is not critical for PBP5-mediated recruitment of SP substrates. As a final test to measure the importance of the D-Ala-D-Ala moiety for PBP5 binding, we performed

both a 2D [¹H,¹⁵N] TROSY NMR-based titration of just the D-Ala-D-Ala peptide with PBP5 (SP3, Fig. 4b). These experiments showed that no CSPs were observed, even at 1:1000 ratios of PBP5:D-Ala-D-Ala (Fig. 4h; ratios ~12.5-fold higher than the highest ratios tested for SP1 or SP2 with PBP5). These data show that the D-Ala-D-Ala moiety is dispensable for PBP5 recruitment and, furthermore, that any interactions between D-Ala-D-Ala and PBP5 strictly rely on other SP residues/moieties for PBP5 binding.

**Fig. 4 | D-Ala-D-Ala is dispensable for PBP5 recruitment and stem peptides are recruited by PBP5 in the presence of β-lactam antibiotics. a** Experimental design **b** *E. faecium* SP (SP1) and modified peptides (SP2-SP6). Functional building blocks that differ from SP1 are highlighted by a red box with a bold outline (SP4, SP6); SP1 functional building blocks absent in the modified peptides are also absent in the cartoons (SP2, SP3, SP5). **c** Molecular structure of SP1, with yellow circles indicating the [1]H atoms with >50% saturation transfer difference (STD) enhancement in the presence of PBP5. **d** Molecular structure of SP2, with yellow circles indicating the [1]H atoms with >50% STD enhancement enhancement in the presence of PBP5. **e** [1]H/[15]N and **f** [1]H/[13]C ILV methyl CSPs vs residue number plot for PBP5:SP2 (1:80 molar ratio); average+1σ (red line) for the PBP5 TP domain. **g** SP2 CSPs mapped on PBP5 structure (gray surface; PDBid: 6MKA); orange surface: residues with significant changes (underlined residues indicate [1]H/[13]C ILV methyl data). S422 is highlighted in red. **h** Overlay of 2D [1]H,[15]N] TROSY spectrum of PBP5 alone (black) with PBP5:SP3 (pink; 1:1000 molar ratio); SP3 structure shown in beige. **i** Overlay of 2D [1]H,[15]N] TROSY spectrum of PBP5 (black), PBP5 saturated with PenG (PBP5:PenG, 1:8 ratio; green) and in complex with SP1 (PBP5:PenG:SP1, 1:8:80 ratio; pink). **j** [1]H/[15]N and (**k**) [1]H/[13]C ILV methyl CSPs vs residue number plot for PBP5:PenG:SP1 (1:8:80 molar ratio); average+1σ (red line) for the PBP5 TP domain. **l** SP1 CSPs mapped on PBP5:PenG structure (gray surface, PDBid: 6MKG); orange surface: residues with significant changes (underlined residues indicate [1]H/[13]C ILV methyl data). S422 is highlighted in red and PenG is shown as sticks (pink). **m** Molecular structure of SP1, with yellow circles indicating the [1]H atoms with >50% STD enhancement enhancement in the presence of PBP5:PenG.

## *E. faecium* stem peptides are recruited to PBP5 in the presence of β-lactam antibiotics

After establishing that the D-Ala-D-Ala moiety of SP1 does not significantly contribute to PBP5 recruitment, we asked whether SPs can bind PBP5 in the presence of β-lactam antibiotics. To test this, we saturated PBP5 with penicillin (PenG; 1:8 ratio) and then titrated the PBP5:PenG complex with increasing ratios of either SP1 and SP2 and followed the resulting CSPs using 2D [1]H,[15]N] TROSY and 2D [1]H,[13]C] ILV methyl HMQC NMR measurements (Fig. 4i and Supplementary 6a-c). We then compared these CSP measurements with those performed with SP1 and SP2 in the absence of PenG (Figs. 1e–g, 4e–g, 4j–l, Supplementary 6d,e). CSPs observed for SP1 and SP2 in the absence and presence of penicillin were nearly identical. We also performed the reverse 1D [1]H STD NMR experiments with SP1 and SP2 against a saturated PBP5:PenG complex (Supplementary Fig. 6f, g). These data showed that the same SP1 and SP2 atoms (those in D-iGln and L-Lys CH/CH₂ atoms and the 5th D-Ala CHα for SP1) interact with PBP5. These experiments confirm that the recruitment of peptidoglycan precursors by PBP5 involves substrate and enzyme chemical groups that are distal from the D-Ala-D-Ala moiety and the active site serine residue, respectively.

PenG is a small β-lactam antibiotic. Thus, we also saturated PBP5 with an anti-MRSA cephalosporin (ceftaroline; 1:8 ratio), which is significantly larger and binds more extensively in the PBP active site cleft[15], and repeated the NMR measurements with increasing ratios of SP2 (Supplementary Fig. 7a, b). The data showed that the observed CSPs were virtually identical to that observed for both β-lactam free and PenG-saturated PBP5 (Supplementary Fig. 7c, d); i.e., that SP2 binds PBP5 identically both with and without ceftaroline. This was further confirmed by 1D [1]H STD NMR experiment with SP2 with a saturated PBP5:ceftaroline complex (Supplementary Fig. 7e), which showed that SP2 atoms that interact with PBP5:ceftaroline are identical to those that bind β-lactam free PBP5. Taken together, these data highlight that while D-Ala-D-Ala can adopt a conformation similar to the 4-membered ring in β-lactam antibiotics (such as penicillin and ceftaroline), it is not important for PBP5 substrate recruitment, suggesting that the mechanism underlying the efficacy of PBP inactivation by β-lactam antibiotics does not exclusively rely on substrate binding mimicry.

## The L-Lys(D-iAsn) residue of the stem peptide is essential for PBP5 recruitment

The 1D [1]H STD NMR data showed that the L-Lys(D-iAsn) residue of the *E. faecium* SP is critical for PBP5 recruitment. To test this observation, we generated a SP lacking a side chain at the 3rd position by replacing L-Lys(D-iAsn) with Gly (SP4, D-Lac-L-Ala-D-iGln-Gly-D-Ala-D-Ala; Fig. 4b). We then repeated the interaction studies using 2D [1]H,[15]N] TROSY and 2D [1]H,[13]C] ILV methyl HMQC NMR measurements to identify the PBP5 TP domain residues that are affected by SP4 binding (Fig. 5a and Supplementary 8a). In contrast to the results observed with SP1 and SP2 (Fig. 1 and 4), at the highest PBP5-to-SP4 molar ratio (1:80) only a limited number of PBP5 TP domain CSPs were observed (Fig. 5b–d). These data show that SP4 interacts with PBP5 very poorly.

Lastly, we also performed inverse experiments to determine if any elements of SP4 interact with PBP5 (1D [1]H STD NMR; these measurements can identify even weaker interactions than 2D [1]H,[15]N] TROSY and 2D [1]H,[13]C] ILV methyl HMQC NMR measurements). These experiments identified only a residual interaction with the iGln residue and the 5th D-Ala CHα (Fig. 5e and Supplementary Fig. 8b,c). No interactions with Gly were observed. Together, these data confirm that the D-iAsn-substituted L-Lys residue of the SP is critical for PBP5 recruitment.

Due to the chemical shift overlap in the 1D [1]H NMR spectrum of the *E. faecium* SP, it was not possible to directly assess if the D-iAsn group contributes to PBP5 SP binding. To address this, we generated a modified *E. faecium* SP that lacks the D-iAsn modification at the L-lysine side chain that is essential for forming the cross-linking between different *E. faecium* SPs and thus for the formation of the peptidoglycan layer (SP5, D-Lac-L-Ala-D-iGln-L-Lys-D-Ala-D-Ala; Fig. 4b). The 2D [1]H,[15]N] TROSY and 2D [1]H,[13]C] ILV methyl HMQC spectra showed significantly fewer CSPs for SP5 with PBP5 when compared with SP1 or SP2. Furthermore, for those CSPs observed, the extent of the CSP was also reduced, indicative of a weaker interaction (Fig. 5f–i and Supplementary Fig. 9a). Using 1D [1]H STD NMR experiments, we then showed that the L-Lys and D-iGln residues are responsible for the observed weak recruitment (Fig. 5j and Supplementary Fig. 9b,c). Together, these data show that all atoms in the L-Lys(D-iAsn) residue contribute to *E. faecium* SP recruitment to PBP5.

We then tested the contribution of the D-iGln residue for PBP5 binding (D-Lac-L-Ala-D-nGln-L-Lys(D-iAsn)-D-Ala-D-Ala; SP6, Fig. 4b). In this precursor analog, L-Lys is linked to the α-carboxyl of a D-Gln residue rather than to the γ-carboxyl of a D-iGln residue (amidated on the α-carboxyl in SP1 and in authentic *E. faecium* PG precursor). The 2D [1]H,[15]N] TROSY and 2D [1]H,[13]C] ILV methyl HMQC spectra showed CSPs that are again more akin to that of SP1 than SP4 (Supplementary Fig. 10a–e). Using 1D [1]H STD NMR experiments we showed that SP6 has an interaction pattern similar to that of SP1 and includes protons from Cβ in D-iGln (Supplementary Fig. 10f, g). Taken together, these data show that the D-iGln group contributes to binding, but significantly less so than L-Lys(D-iAsn) group.

## Enterococci and related PBPs exhibit broad stem peptide specificities

Finally, it was previously reported that replacing the genes of high molecular weight transpeptidase PBPs with those from other species (i.e., expressing *pbp4* from *E. faecalis* instead of *pbp5* in *E. faecium*) had little impact on β-lactam resistance, suggesting they exhibit broad substrate specificities (i.e., PBP4$_{E.faecalis}$ can cross-link branched peptides containing the D-iAsn sidechain present in *E. faecium* rather than the L-Ala-L-Ala side-chain present in its endogenous *E. faecalis* host)[7]. Our data show that SP recruitment is primarily due to the L-Lys residue, which is conserved between the two hosts, suggesting that the observed cross-reactivity of organism-specific PBPs for different PG is due, in part, to the conservation of the L-Lys residue. Thus, we reasoned that PBP4$_{E.faecalis}$ and PBP2B$_{E.faecalis}$ should effectively recognize

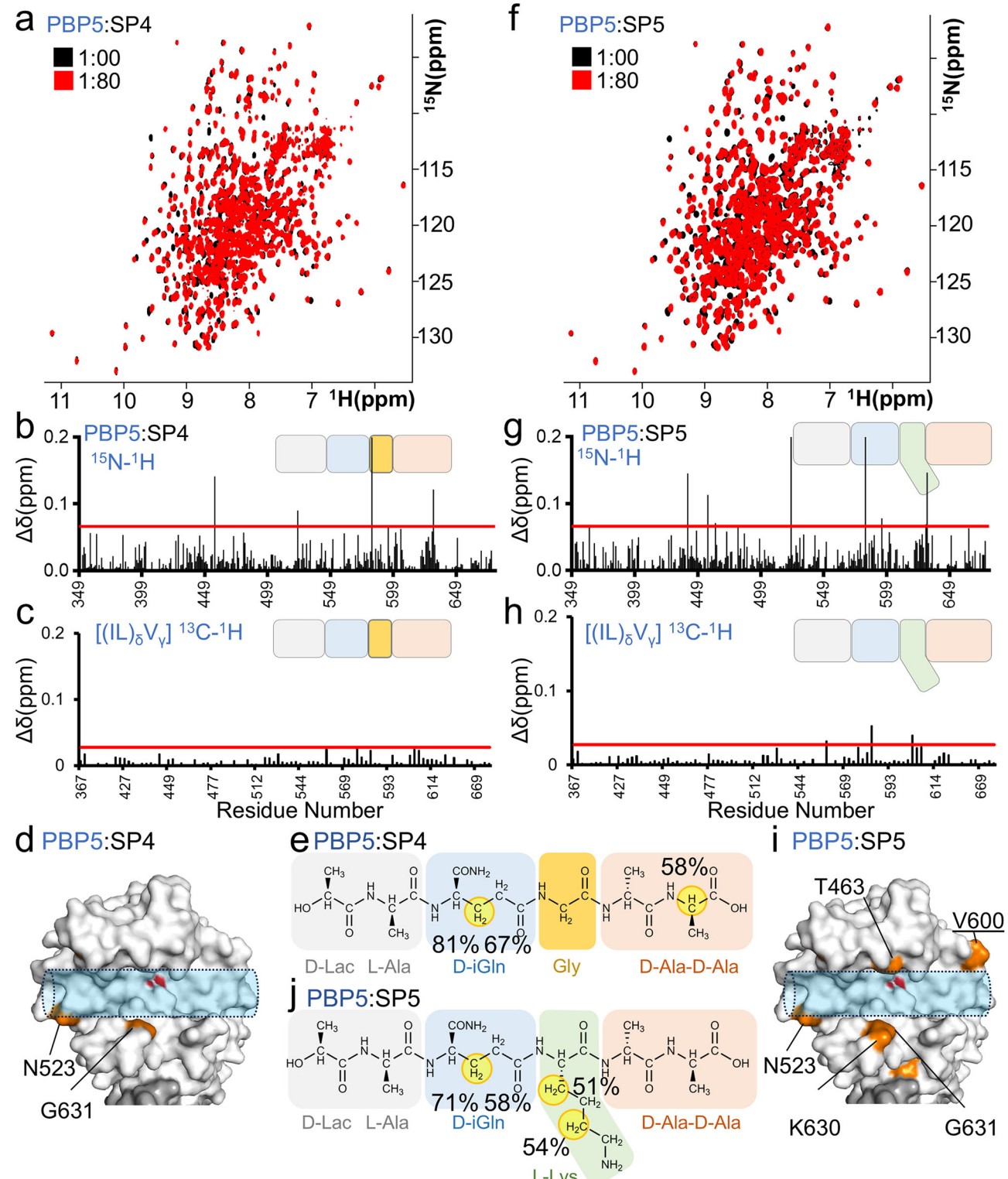

**Fig. 5 | The L-Lys(D-iAsn) residue is necessary and sufficient for PBP5 binding.**
**a** Overlay of 2D [¹H,¹⁵N] TROSY spectrum of PBP5 (black) with SP4 (1:80 ratio; red).
**b** ¹H/¹⁵N and **c** ¹H/¹³C ILV methyl CSPs vs residue number plot for PBP5:SP4 (1:80 ratio); average+1σ (red line) for the PBP5 TP domain. **d** SP4 CSPs mapped on PBP5 structure (gray surface, PDBid: 6MKA); orange surface: residues with significant changes (underlined residues indicate ¹H/¹³C ILV methyl data). S422 is highlighted in red. **e** Molecular structure of SP4, with yellow circles indicating the ¹H atoms with >50% saturation transfer difference (STD) enhancement. **f** Overlay of 2D

[¹H,¹⁵N] TROSY spectrum of PBP5 (black) with SP5 (1:80 ratio; red). **g** ¹H/¹⁵N and **h** ¹H/¹³C ILV methyl CSPs vs residue number plot for PBP5:SP5 (1:80 ratio); average +1σ (red line) for the PBP5 TP domain. **i** SP5 CSPs mapped on PBP5 structure (gray surface, PDBid: 6MKA); orange surface: residues with significant changes (underlined residues indicate ¹H/¹³C ILV methyl data). S422 is highlighted in red. **j** Molecular structure of SP5, with yellow circles indicating the ¹H atoms with >50% saturation transfer difference (STD) enhancement.

the precursor from *E. faecium*. To test this hypothesis, we performed 1D $^1$H STD NMR experiments with the *E. faecium* SP2 SP with PBP4$_{E.faecalis}$ and PBP2B$_{E.faecalis}$ (Fig. 6a, b and Supplementary Fig. 11a, b). The STD experiments showed the same elements of SP2 interact with PBP4$_{E.faecalis}$ and PBP2B$_{E.faecalis}$ as with PBP5 from *E. faecium*, confirming the central importance of the L-Lys residue for PBP recruitment and defining the molecular basis of the broad substrate specificities exhibited by high molecular weight PBPs.

## Discussion

The peptidoglycan wall is essential for bacterial survival. Peptidoglycan building blocks are extensively cross-linked via transglycosylation and transpeptidation to provide the mechanical strength to sustain the turgor pressure of the cytoplasm[2]. Thus, it is not a surprise that blocking these biosynthetic steps, i.e., by inhibiting transpeptidation with β-lactam antibiotics like penicillin, negatively compromise bacterial viability. An orthogonal strategy to block transpeptidation with high barriers to resistance is to block the ability of PG/SPs to bind PBPs using non-covalent inhibitors that target PG/SP binding sites. However, this strategy has remained unexplored as a detailed molecular understanding of how PG is recruited to PBPs has been lacking. This knowledge gap is due, in part, to the difficulties in preparing homogeneous PG preparations and the weak affinity of PBPs for peptidoglycan precursor analogs that have prevented crystallization or cryo-EM structure analysis of PBPs in complex with their substrates. Here we overcame these limitations by employing biomolecular NMR spectroscopy coupled with sophisticated SP synthetic methods to decipher the mode of recruitment of PG and SPs by PBPs and compare the modes of binding of peptidoglycan precursors and β-lactams. Although PBPs are large single-chain proteins, and thus extremely challenging targets for NMR spectroscopy, extensive optimization[11] allowed us to identify conditions suitable for the NMR analysis of *E. faecium* PBP5 and its interactions both with disaccharide-peptide fragments of its endogenous PG and with a variety of synthetic peptides that are derivatives of various components of the PG SPs.

Our data identified the PBP5 residues that serve as the core PG recruitment sites. Instead of being located immediately surrounding the catalytic nucleophile (S422 in PBP5), the main interacting residues are located at the outer edges of the active site cleft of the enzyme (Figs. 1 and 2). Because most PBP5 resistance variants contain substitutions that occur above or below (versus on the outer edges) of the catalytic cleft[11], our identification of these PG recruiting sites on PBP5 explains why PBP5 variants that are resistant to β-lactams can still perform transpeptidation, i.e., the resistance substitutions are distal from the SP binding sites and thus do not affect SP recruitment. Using *E. faecium* survival assays, we also showed that the experimentally determined PG recruitment sites on PBP5 are not only physiologically relevant, but essential for bacterial viability as, unlike wt PBP5, PBP5 variants with mutated recruitment sites fail to rescue *E. faecium* growth (Fig. 3). The identification of the peptidoglycan recruitment sites on PBP5 and our demonstration that altering these sites inhibit *E. faecium* growth suggest that antibiotics that target these sites are likely to exhibit high barriers to resistance.

We also discovered that the recognition of SPs by PBP5 is not achieved via extensive contacts throughout the *E. faecium* SP, but instead by interactions restricted to the 3rd position, L-Lys(D-iAsn), with additional small contributions from the D-iGln residue at the 2nd position (Figs. 4 and 5). Consistent with this result, we determined that the SP interaction with PBP5 is weak (~4 mM), with the recruitment of reduced PG being comparatively enhanced, suggesting that the additional elements present in PG, such as the disaccharide moiety, contribute to PG precursor binding. Unexpectedly, these data are inconsistent the hypothesis that β-lactam efficacy is due, in part, to its ability to mimic the D-Ala-D-Ala C-terminal moiety of SPs. Our data show, instead, that the major recognition element is the L-Lys(D-iAsn)

residue, with the D-Ala-D-Ala being dispensable for binding (D-Ala-D-Ala may still have a critical role in the catalytic reaction as we have not tested its role in the transition state). This result was confirmed with experiments with just the D-Ala-D-Ala moiety, which failed to interact with PBP5, and an *E. faecium* SP derivative that lacked the D-Ala-D-Ala moiety, which bound PBP5 in a manner nearly identical to the native SP (Fig. 5). Furthermore, we showed that SPs bind PBP5-β-lactam adducts demonstrating that SP binding and PBP5-β-lactam adduct formation are not mutually exclusive (Fig. 4). Thus, these data show that while D-Ala-D-Ala and β-lactams adopt similar conformations, they clearly engage PBPs via distinct mechanisms. Finally, these data also explain why PBPs from different species can recruit the same SP following PBP exchange between bacteria[7], i.e., the key residue that mediates recognition, L-Lys, is conserved in Gram-positive cocci (Fig. 6a, b).

Together, these comprehensive data provide key molecular insights into the binding and processing of PG by PBPs (Fig. 6c) and explain why clinically relevant β-lactam resistant variants can still process SPs to form PG, i.e., mutations that affect β-lactam antibiotic-mediate inhibition do not alter PG recruitment. This knowledge unlocks different avenues for the rational design of drugs that will block the interaction of SPs with PBPs, which, in turn will prevent crosslinking ultimately resulting in bacterial death (Fig. 6d). Importantly, our demonstration that mutations in the PG recruitment site inhibit bacterial growth strongly suggest that antibiotics targeting this site will also have high barriers to resistance, since mutations that disrupt antibiotic binding will also disrupt SP recruitment.

## Methods

### Plasmid construction

PBP5$_{37-678}$ from *Enterococcus faecium*, PBP4$_{36-680}$ and PBP2B$_{56-711}$ from *E. faecalis* were cloned into the pRP1B vector with His$_6$-tag and a Tobacco Etch Virus (TEV) protease cleavage site as previously described[15].

### Protein Expression

PBP2B, PBP4 and PBP5 were expressed and purified as described earlier[15]. Briefly, the plasmid DNAs were transformed into *Escherichia coli* BL21 (DE3) cells (Agilent). Cells were grown in LB medium in the presence of selective antibiotic (kanamycin) at 37 °C until an OD$_{600}$ ~ 0.8–1.1, upon when the cells were cooled down to 18 °C prior to the addition of 0.5 mM isopropyl β-D-thiogalactopyranoside (IPTG; GoldBio) for protein expression. Cells were harvested 18–20 h after induction by centrifugation (8000 × $g$ for 15 min; 4 °C). Pellets were stored at −80 °C until purification.

Uniformly ($^2$H,$^{15}$N)-labeled PBP5 was achieved by growing cells in D$_2$O based M9 minimal media containing 1 g/L $^{15}$NH$_4$Cl and/or 4 g/L ($^2$H,$^{13}$C)-D-glucose (CIL or Sigma-Aldrich) as the sole nitrogen and carbon sources, respectively. Furthermore, a uniformly ($^2$H,$^{15}$N)-labeled PBP5 with Ile, Leu, and Val selectively labeled with $^{13}$C methyl for titrations was achieved by growing cells in D$_2$O based M9 minimal media containing 1 g/L $^{15}$NH$_4$Cl and/or 4 g/liter ($^2$H,$^{12}$C)-D-glucose, 120 mg/liter ($^{13}$C$_5$,3-$^2$H$_1$) α-ketoisovaleric acid (CDLM 7317) and 60 mg/liter ($^{13}$C$_4$,3,3-$^2$H$_2$) α-ketobutyric acid (CDLM 7318) in 100% D$_2$O as the sole nitrogen and carbon sources, respectively[16]. Multiple rounds (0%, 25%, 50%, 75% and 100%) of D$_2$O adaptation were necessary for high-yield expression.

To rigorously verify the sequence-specific backbone assignment of PBP5, we produced uniformly ($^2$H,$^{15}$N)-labeled PBP5 variants E269A, G458A, E494A, E508A, E524A, D525A, N528A, D530A, Q540A, E542A, E583A, E622A and E625A. Furthermore, we produced $^{15}$N valine selective labeled samples of PBP5 V468A, V556A and V596A; a $^{15}$N tyrosine selective labeled sample of PBP5 Y460A; a $^{15}$N phenylalanine selective labeled sample of PBP5 F557A and $^{15}$N leucine selective labeled samples of PBP5 L607A and L610A. Selective labeling of Phe, Tyr, Leu or Val residues was performed as previously described[11]. In brief, $^{15}$N-Phe,

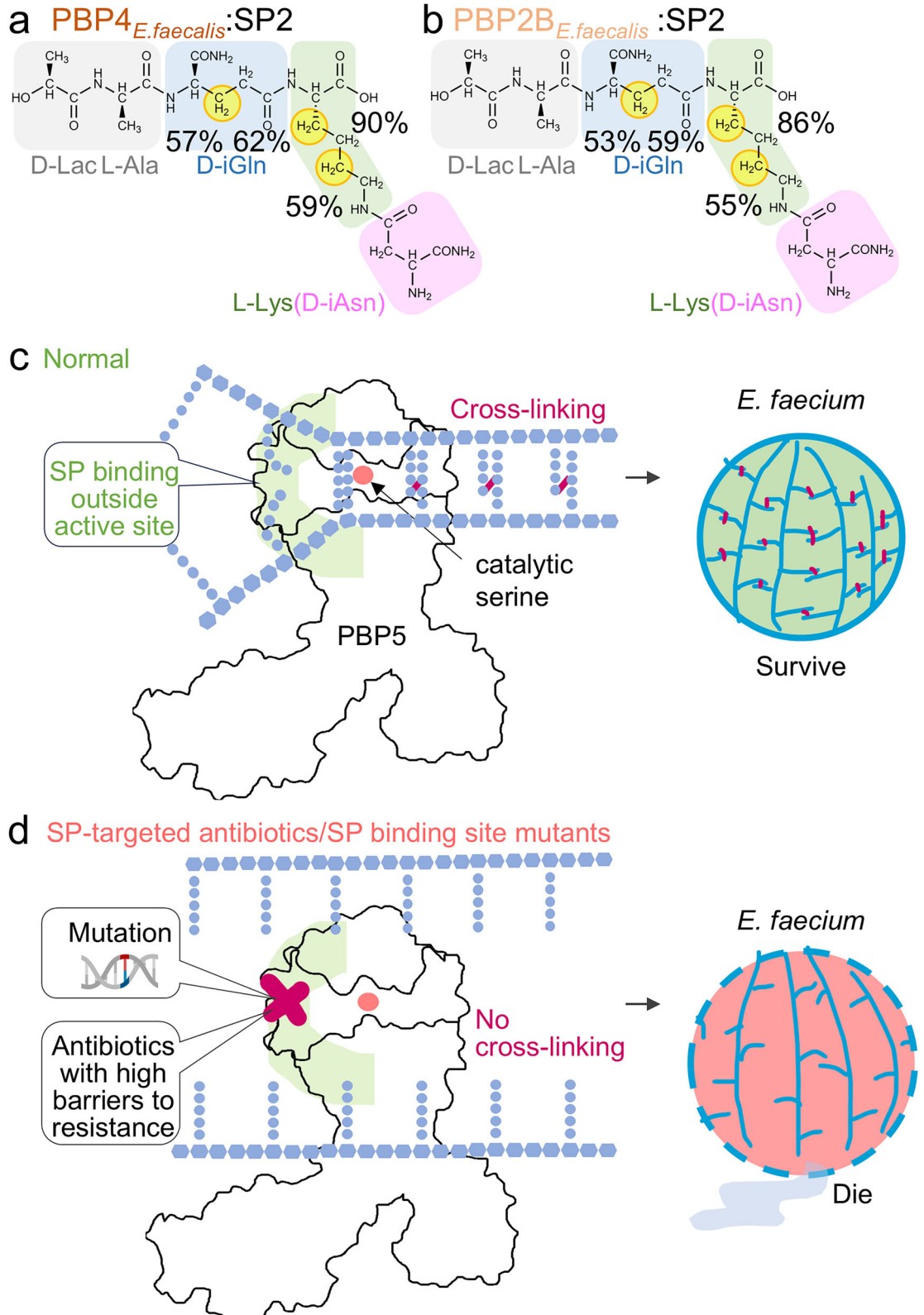

**Fig. 6 | Distal PG recruitment to PBP5 allows for new antibiotic design possibilities. a** Molecular structure of SP2, yellow circles highlight [1]H atoms with >50% STD enhancement when in complex with PBP4 of *E. faecalis*. **b** Molecular structure of SP2, with yellow circles indicating the [1]H atoms with >50% STD enhancement when in complex with PBP2B of *E. faecalis*. **c** The SP recruitment site on PBP5 is distal from the active site and catalytic serine, with SP recruitment leading to efficient transpeptidation and formation of the peptidoglycan cross-links that results in robust bacterial growth. **d** Mutations in the SP recruitment pocket are not viable, suggesting that antibiotics targeting this interaction pocket will have high barriers to resistance as mutations that disrupt antibiotic binding will also likely negatively impact SP recruitment, which will also be nonviable.

$^{15}$N-Tyr, $^{15}$N-Leu or $^{15}$N-Val amino acid was added with unlabeled L-amino acids to M9 media[17].

## Purification

Cell pellets were resuspended in Lysis Buffer (50 mM Tris pH 8.0, 500 mM NaCl, 5 mM imidazole, 0.1% Triton X-100) and lysed using high-pressure homogenization (EmulsiFlex C3; Avestin). The cell lysate was centrifuged at 40,000 × g for 45 min at 4 °C. The clarified lysate was filtered through 0.22 μm PES filter (Millipore) and loaded onto a pre-equilibrated Ni-NTA gravity column (Genesee Scientific), washed with 100 mL His-tag Buffer A (50 mM Tris pH 8.0, 500 mM NaCl, 5 mM imidazole), before being eluted with 60% His-tag buffer B (50 mM Tris pH 8.0, 500 mM NaCl, 250 mM imidazole). The protein was pooled and dialyzed with TEV protease (produced in house) for 48–60 h at 4 °C to cleave the His$_6$-tag. A subtraction purification was carried out using a Ni-NTA gravity column to remove the His$_6$-tag and TEV. PBP2B, PBP4 or PBP5 in the flow-through was collected and dialyzed for at least 5 h into 10 mM Tris pH 8.0, 1.5 M (NH$_4$)$_2$SO$_4$. The dialyzed protein was loaded onto two 5 ml pre-equilibrated HiTrap Phenyl HP hydrophobic interaction columns (Cytiva) and a gradient of decreasing (NH$_4$)$_2$SO$_4$ was used to separate the folded and misfolded PBP proteins. Properly folded PBPs (first peak) were collected and immediately dialyzed against NMR buffer (10 mM MES pH 5.8, 25 mM NaCl). Dialyzed PBP2B, PBP4 or PBP5 was concentrated and loaded onto an NMR buffer pre-equilibrated SEC 200 26/60 column (Cytiva) for the final purification. The fractions corresponded to PBP2B, PBP4 or PBP5 were pooled and concentrated for NMR measurements or storage by snap freezing in liquid nitrogen and kept at −80 °C.

## Protein refolding

-100 expected peaks were missing in the 2D [$^1$H,$^{15}$N] TROSY spectrum of PBP5 due to a lack of D/H exchange caused by expression of PBP5 in D$_2$O-based medium. For PBP5 refolding, purified PBP5 was diluted to 0.1–0.2 mg/mL in Refolding Buffer (10 mM Tris pH 8.0, 800 mM NaCl) and placed in a SnakeSkin dialysis tubing (3 kDa MWCO; ThermoFisher Scientific). The dialysis tube was then placed into Refolding Buffer containing 6 M GuHCl overnight for denaturation. The next day, the tube was transferred to Refolding Buffer with 3 M GuHCl. After 24 h, the tube was transferred to Refolding Buffer with 1.5 M GuHCl for 4 h, 0.5 M GuHCl for 4 h and Refolding Buffer with no GuHCl for 4 h. Finally, the protein was dialyzed into 10 mM Tris pH 8.0, 1.5 M (NH$_4$)$_2$SO$_4$ for HIC purification, followed by SEC in NMR buffer, as described above.

## Synthesis of stem peptides

All reagents were obtained from commercial suppliers and used without further purification. Low resolution mass spectra (LRMS) were obtained on an LCQ Advantage mass spectrometer (ThermoFisher Scientific) under electrospray ionization in positive or negative mode. Peptides were assembled using an ABI 433A Peptide synthesizer (Applied bioscience). Flash chromatography purifications were performed on a Companion system, and rpHPLC analysis were performed on an UltiMate 3000 system (ThermoFisher Scientific).

## Peptidoglycan extraction

The peptidoglycan was extracted using the hot SDS method as described previously[7] with minor modifications. *Enterococcus faecium* strain D344R was grown in 4 L of brain heart infusion broth at 37 °C to an optical density at 600 nm of 0.8. The following treatments were applied in parallel for four 1 L aliquots of the culture. Bacteria were collected by centrifugation (7500 × g for 15 min at 4 °C). They were resuspended in 20 ml of 4% SDS prewarmed at 100 °C and incubated for 30 min at 100 °C. Peptidoglycan was collected and washed five times with water (20 ml) by centrifugation (53,000 × g for 10 min at 25 °C). Peptidoglycan was treated overnight at 37 °C with 200 μg/mL pronase (Sigma) in 1 mL of 10 mM Tris-HCl pH 7.5. Following the

pronase treatment, peptidoglycan was washed two times with water (20 mL) by centrifugation (53,000 × g for 10 min at 25 °C). Peptidoglycan was then treated with 200 μg/mL trypsin (Sigma) in 1 mL of phosphate buffer (20 mM pH 8.0). Peptidoglycan was again collected by centrifugation, resuspended in 4% SDS at 100 °C, and incubated for 30 min at 100 °C. Peptidoglycan was washed five times with 20 mL of water and then treated with 200 μg/mL mutanolysin (Sigma) and 200 μg/ml lysozyme (Sigma) for 18 h at 37 °C in 1 mL of 50 mM Tris-HCl pH 8.0. Soluble disaccharide-peptides were separated in 5 tubes (0.2 mL) and reduced by the addition of 0.2 mL of 250 mM borate buffer pH 9.0 containing 10 mg/mL sodium borohydride. The solution was incubated for 30 min at 25 °C and insoluble material was removed by centrifugation (12,000 × g for 5 min at 25 °C). Muropeptides were lyophilized and stored at −20 °C.

## Synthesis of non-commercial precursors for peptide synthesis

1. Synthesis of Boc-D-Asp-NH$_2$

Ethyl chloroformate (0.30 mL, 3.40 mmol) was added dropwise at −20 °C to a solution of Boc-D-Asp(OBn)-OH (1 g, 3.1 mmol, Sigma-Aldrich) and triethylamine (0.46 mL, 3.40 mmol) in anhydrous THF (30 mL). The solution was stirred for 30 min. NH$_4$OH (28%, 1 mL, 15.5 mmol) was added at 0 °C and the reaction mixture was stirred at 0 °C for 2 h. The solvent was removed under reduced pressure, and the residue was dissolved in dichloromethane. The organic phase was washed with water, dried over MgSO$_4$, filtered and concentrated, affording Boc-D-Asp(OBn)-NH$_2$ as a white powder (1 g, 91%) that was used without further purification. To a solution of Boc-D-Asp(OBn)-NH$_2$ (1 g, 3.1 mmol) in anhydrous THF (30 mL) was added 10 wt. % Pd/C (100 mg). The reaction mixture was hydrogenated overnight under atmospheric pressure. The solution was filtered through celite and THF was evaporated under reduced pressure. Flash chromatography (Dichloromethane/MeOH, 9:1) afforded Boc-D-Asp-NH$_2$ 1 as a white powder (0.56 g, 78%). LRMS: Calculated for C$_9$H$_{15}$N$_2$O$_5$ [M-H]$^-$: 231.1; measured: 231.0. $^1$H and $^{13}$C NMR spectroscopic data were consistent with previously published results[12].

2. Synthesis of Fmoc-L-Ala-D-iGln

H-Ala-D-iGln from Bachem (2 g, 9.2 mmol) was dissolved in water/acetone solution (100 mL, 1/1) and cooled on an ice bath before addition of NaHCO$_3$ (1.55 g, 18.4 mmol). Fmoc-Cl (2.86 g, 11.04 mmol) was dissolved in acetone (50 mL) and slowly added to the solution. After the addition, the reaction mixture containing a white precipitate was allowed to warm to room temperature and was left under stirring for 4 h. Acetone and water were removed under reduced pressure. Crude product was dissolved in glacial acetic acid (200 mL) and purified by flash chromatography on a C18 column (0.1%TFA to ACN/0.1%TFA). Product 2 was identified by mass spectroscopy and analyzed by rp-HPLC (gradient from 0 to 100% ACN in 0.1% TFA applied between 10 and 40 min) at 1 ml.min$^{-1}$. RT = 27.7 min. Pure product 2 was lyophilized as a white powder (2.4 g; 59% yield). LRMS: Calculated for C$_{23}$H$_{26}$N$_3$O$_6$ [M + H]$^+$: 440.2; measured 440.1 (Supplementary Fig. 12, Supplementary Table 4).

3. Synthesis of D-lactic acid derivative

A solution of D-lactic acid-benzyl ester (5.0 g, 27.74 mmol) in dry DCM (18 mL) was treated with di-t-butyl decarbonate (15.14 g, 69.40 mmol) and magnesium perchlorate (0.63 g, 2.80 mmol). The mixture was refluxed for 12 h. The solution was poured into a saturated aqueous NaHCO₃ solution, and the mixture was extracted with EtOAc. The organic layer was washed with water, dried with brine and over MgSO₄, filtered, and concentrated under reduced pressure. The residue was purified by flash chromatography (silica gel, cyclohexane/AcOEt 75/25) affording O-tBu-D-lactic acid benzyl ester as a colorless oil (3.5 g, 54%). O-tBu-D-lactic acid benzyl ester (3.5 g, 7.62 mmol) was dissolved in MeOH (75 mL) and treated with 10 wt. % Pd/C (0.5 g) under 4 bar of H₂ for 12 h at room temperature. The residue was filtered through celite and concentrated under reduced pressure to afford D-lactic acid derivative 3 as a colorless oil (1.00 g, 90%). $^1$H and $^{13}$C NMR spectroscopic data were consistent with previously published results[12]. LRMS: Calculated for $C_7H_{15}O_3^+$ [M + H]$^+$: 147.1; measured: 146.9 (Supplementary Fig. 12, Supplementary Table 4).

*Solid-phase synthesis of linear peptides (SP4 and SP5)*

SP4

SP5

Peptides were assembled on an ABI 433A Peptide synthesizer (Applied bioscience) using the Fastmoc protocol, Fmoc-D-Ala-Wang resin as a first amino acid (0.80 mmol.g⁻¹, 125 mg, BACHEM) and following the users manual with minor modifications. Briefly, a molar equivalent excess of 10 or 3 of the commercial and non-commercial Fmoc-amino acids was used during coupling, respectively. The Fmoc-amino acids (or Fmoc-L-Ala-D-iGln) were dissolved and activated in cartridge in a mixture of 2.0 g of 0.45 M HBTU/HOBt in DMF, 2 M DIEA, and 0.8 mL NMP. Amine deprotection was performed in NMP containing 20% piperidine. Final deprotection of acid labile protecting groups and cleavage from the resin were carried out under agitation at room temperature for 2 h with 10 mL of a TFA solution containing DCM, TIPS and water (80:20:5:5 v/v/v/v). The solution was filtered, and the resin was washed with 1 mL of TFA. The TFA solutions were pooled and TFA evaporated by nitrogen bubbling. The crude product was dissolved in a 10% acetic acid solution (5 mL) and extracted with chloroform (3 × 10 mL). The aqueous solution was lyophilized, and the peptides were purified by *rp*HPLC on a 5 μm Nucleosil preparative C-18 column (22 × 250 mm) using a linear gradient (0 to 100% buffer B) applied between 10 min and 40 min (Buffer A: 0.1 % TFA in H₂O; buffer B: 0.1 % TFA in CH₃CN; 10 mL.min⁻¹). Peptide-containing fractions were identified at 214 nm and analyzed by mass spectrometry. Purity was assessed by *rp*HPLC (analytical C18 Nucleosil column, 3 μm, 4.6 × 250 mm) using a linear gradient (0 to 100 buffer B) applied between 10 and 40 min (Buffer A: 0.1 % TFA in H₂O; buffer B: 0.1 % TFA in CH₃CN; 1 mL.min⁻¹) (Supplementary Fig. 12, Supplementary Table 4).

*Solid-phase synthesis of branched peptides (SP1, SP2, and SP6)*

Branched peptides were synthesized via divergent Fmoc solid phase peptide synthesis using orthogonal protecting groups on the lysine residue. The starting material for SP1 and SP6 was commercial Fmoc-D-Ala-Wang resin (BACHEM). For SP2 peptide, the starting material was commercial Wang resin (BACHEM) loaded manually with L-Lys containing both α and ε NH₂ groups protected with Fmoc and ivDde groups, respectively. Peptides were assembled on a ABI 433A peptide synthesizer on the resin (0.69 mmol.g⁻¹, 75 mg) as described above to afford Ot-Bu-D-Lac-L-Ala-D-iGln-L-Lys(ivDde)-D-Ala-D-Ala-WANG resin, Ot-Bu-D-Lac-L-Ala-D-iGln-L-Lys(ivDde)-WANG resin, and Ot-Bu-D-Lac-L-Ala-D-nGln-L-Lys(ivDde)-D-Ala-D-Ala-WANG resin. For each peptide, the ε NH₂ group of L-Lys was deprotected manually by hydrazinolysis of the ivDde protective group by a 5% of hydrazine mono-hydrate solution in DMF (3 × 15 min). The resin was washed with DCM (5 × 10 mL) and DMF (5 × 10 mL) and was reintroduced in the ABI 433A Peptide synthesizer for final incorporation of the Boc-D-*iso*Asn. Final deprotections, purifications, and purity assessments were performed as described above (Supplementary Fig. 12, Supplementary Table 4).

## Synthetic gene design for disk diffusion assays

The pbp5 sequence from *E. faecium* D366 was used as a template to create the synthetic genes pbp5_V1 to pbp5_V6 (pbp5_V1: N523A; pbp5_V2: N523A_Q627A; pbp5_V3: loop 625-631; pbp5_V4: D530A_L533A_W457A; pbp5_V5: Q396A_E397A; pbp5_V6: E284A_E291A_S294A_N295A). The synthetic genes were purchased from GeneWiz (South Plainfield, NJ). The ribosome binding site sequence AGGAGG (Shine-Dalgarno), and a string of five A's followed by the sequence ATGAAAAGAAGTGAC, that corresponds to the first 15 bp found in *pbp5* from *E. faecium* C68 were added upstream of the START codon. The terminator sequence GAACGACTTCATAACTATGTAAAAGGACTGTGAC GGTATCGTCACAGTCTTTTTT was added downstream of the STOP codon. BamHI restriction sites were added to the flanking 5′ and 3′ ends. The synthetic genes were inserted into the vector pCWR624Δpbp5. This vector was derived from pCWR624[10], in which the *pbp5* gene from D366 was excised by digestion with BamHI followed by re-ligation of the vector, allowing subsequent insertion of modified synthetic pbp5s into the BamHI site downstream of *ftsW*$_{Efm}$ and *psr* from *E. faecium* C68[10]. The synthetic genes were inserted in a BamHI site located downstream of *psr*. The proper orientation was confirmed by PCR amplification using the forward primer OR_pbp5_FOR_1 (5′-CAAGCGAGAGGTTCTTTTAG-3′) and the reverse primer OR_pbp5_REV_1 (5′-GGACAAATTGCTCAACTGTC-3′). Plasmids were sequence verified for the specific binding site mutations. The resulting constructions were electroporated into the natural pbp5(-) mutant *E. faecium* D344SRF with a selection on brain heart infusion agar (BHI) plates containing kanamycin (1500 μg/ml).

SP1

SP2

SP6

**Disk diffusion assay and MIC determination by micro broth dilution method**

*E. faecium* strains were streaked fresh from −80 °C stored glycerol stocks onto brain heart infusion (BHI) agar plates and incubated for 18 h at 37 °C. Strains harboring the plasmid derivatives were grown under antibiotic pressure on agar plates containing kanamycin at 1500 μg/ml. A single colony was inoculated into BHI or BHI-kanamycin (1500 μg/ml) for overnight cultures (18 h).

Antibiotic disks used were manufactured by BD BBL™. Disk diffusion assays were performed according to the Clinical and Laboratory Standards Institute (CLSI) guidance. Overnight cultures were inoculated into 3 ml of BHI or BHI-kanamycin (1500 μg/ml) broth with a dilution of 1:100 and incubated until an $OD_{600}$ of ~0.5 was reached. The $OD_{600}$ was adjusted to 0.01 to get the inoculum suspension. Regular BHI agar plates or supplemented with kanamycin (1500 μg/ml) were inoculated by swab for a lawn of bacteria and the antibiotic susceptibility disks Ampicillin (10 μg), Penicillin (10 U) and Ceftriaxone (30 μg) were placed onto the inoculated plates. The plates were incubated at 37 °C and the zones of inhibition were measured after 24 h. All experiments were conducted in triplicate.

MIC determination was performed using a standard broth microdilution technique in duplicate from independent cultures[18]. Overnight cultures were inoculated into 3 ml of BHI or BHI-kanamycin (1500 μg/ml) broth in a dilution of 1:100 and incubated until an $OD_{600}$ of ~0.5 was reached. Cultures were diluted 1:1000 and inoculated into the 96-well plates with the corresponding antibiotics to be tested. Plates were incubated at 37 °C for 20 h and read by eye. The MICs were defined as the lowest antibiotic concentration that inhibited growth.

**Western blot for PBP5 expression**

Cell lysates were prepared from 40 mL cultures of mid-log phase cell growths and processed for PBP5 expression by western blot as previously described[19]. In brief, cells were processed by bead-beating and proteins separated by electrophoresis in 10% Bis-Tris NuPAGE gels (ThermoFisher Scientific), with transfer to PVDF membranes (Invitrogen). Immunodetection was performed using custom polyclonal chicken anti-PBP5 antibody provided by New England Peptide, Inc. Gardner, MA. The primary antibody was diluted 1:4000 and secondary Goat anti-chicken IgY (H&L) HRP 1:10,000 (ThermoFisher Scientific, #A16054). Clarity ECL (BioRad) was used for detection. Protein transfer to the blot was visualized using Coomassie blue R250 to confirm equivalent loading.

**Bocillin binding competition assay**

The assays were performed in low-volume 384-well black polystyrene assay plate (Corning, 4511), using assay buffer (100 mM Na-phosphate pH 7.0, 0.01% Triton X-100) to prevent the adsorption of Bocillin FL (Invitrogen, B13233) to the plate. Bocillin FL was dissolved in DMSO and stored as a 1 mM stock in the −20 °C freezer. Fresh working stock of Bocillin FL was made each day for the assays by diluting the stock with the assay buffer for a final concentration of 30 nM. The final concentration of PBP5 for the assays was 3.6 μM. The antibiotic, either penicillin G, ceftaroline or ceftriaxone, was added to PBP5 before the reactions were initiated by the addition of Bocillin FL. The plate was sealed with ThermalSeal RT silicone adhesive film (EXCEL Scientific, TSS-RTQ-100) to prevent evaporation during the measurements. Reaction without PBP5 (Bocillin FL only) was used as negative control. All reactions were carried out at 37 °C in a CLARIOStar plate reader (BMG Labtech) equipped with the 482 nm excitation filter, 530 nm emission filter and a dichroic filter LP504. The focal height was adjusted to 11.3 mm and the gain was set to 10 mP using the negative control. Measurements were made at 40 s intervals for up to 75 min with 100 flashes per measurement. Three replicates were made for each treatment in each run, and the experiment was repeated 3 times.

**NMR spectroscopy**

All NMR data were collected on Bruker Avance Neo 600 MHz or 800 MHz spectrometers equipped with TCI HCN Z-gradient cryoprobes at 308 K. NMR measurements of PBP5 were recorded using either ($^2$H,$^{15}$N) or ($^2$H,$^{15}$N)/$^{13}$C ILV labeled PBP5 at a final concentration of 0.1-0.5 mM in NMR buffer (10 mM MES pH 5.8, 25 mM NaCl) and 90% $H_2O$/10% $D_2O$. Data was processed using Topspin 4.1.4 (Bruker) and analyzed using NMRFAM-SPARKY 1.3 and Poky (build number 20240829)[20,21]. The sequence-specific backbone assignment of PBP5 is publicly available at BMRBID 51690.

**NMR analysis of stem peptide interaction**

SPs (SP1 to SP7) were titrated into 0.5 mM ($^2$H,$^{15}$N)-labeled and 0.1 mM ($^2$H,$^{15}$N)/$^{13}$C ILV labeled PBP5 at molar ratios of 0:1, 5:1, 10:1, 20:1, 40:1, 60:1, and 80:1 (stem peptide/SP:PBP5; molar ratios). 2D [$^1$H,$^{15}$N] TROSY or 2D [$^1$H,$^{13}$C] HMQC spectra were recorded for each titration point. SPs were solubilized in NMR buffer. Chemical shift perturbations (Δδ) between PBP5 and SP bound PBP5 spectra were calculated using Eq. 1,

$$\Delta\delta(ppm) = \sqrt{\left(\Delta\delta_H\right)^2 + \left(\frac{\Delta\delta_X}{Y}\right)^2} \quad (1)$$

where, $X$ is $^{15}$N or $^{13}$C and $Y$ is the constant i.e., 10 for $^{15}$N and 4 for $^{13}$C.

Residues with significant CSPs were identified by calculating the standard deviation of all CSPs ($\sigma$), then excluding any CSPs larger than average+$2\sigma$, recalculating $\sigma$ as a new corrected $\sigma_0$, and then repetitively removing CSPs above $2\sigma_0$ until no additional CSPs can be removed. CSPs above $\geq 2\sigma_2$ was the statistical threshold for $^1$H/$^{15}$N backbone CSP analysis and $\geq 2x\sigma_1$ for $^1$H/$^{13}$C methyl ILV data for the interaction of SP1 with PBP5. The same threshold values were used for all other peptides to allow for direct comparison with the SP1/PBP5 interaction. The identical significance analysis was also performed for PenG/ceftaroline saturated PBP5 (1:8 molar ratio of PBP5 to β-lactam antibiotics was used to ensure saturation of an extended time period needed for the SP titrations) SP interaction studies; CSPs above $\geq 2\sigma_3$ was the statistical threshold for $^1$H/$^{15}$N backbone CSP analysis and $\geq 2x\sigma_2$ for $^1$H/$^{13}$C methyl ILV data to identify all residues with a statistically significant change in its chemical shift upon SP interaction[22].

Residue specific $K_D$ values were calculated by using Eq. 2[23],

$$CSP_{obs} = \frac{CSP_{max}}{2[P]}\left( ([P] + [S] + K_D) - \sqrt{([P] + [S] + K_D)^2 - 4[S][P]} \right) \quad (2)$$

where $P$ and $S$ are the concentration of PBP5 and SP1, respectively. $CSP_{obs}$ is the experimentally observed chemical shift value and $CSP_{max}$ is the largest experimentally observed chemical shift value of the residue. In our analysis specifically, the reported $K_D$ values are overestimated because our titrations even at the highest peptide to protein ratios (e.g., 1:80 for PBP5:SP1 molar ratio) did not reach saturation. The lack of saturation was due to the limited amount of available SP peptide due to their difficult synthesis. Residues used for $K_D$ calculation were common residues with statistically significant CSPs in the interaction studies of SP1 and SP2 with PBP5 at a 1:80 ratio.

**STD (saturation transfer difference) NMR experiments**

STD NMR experiments were carried out on 0.02 mM unlabeled PBP5$_{T485}$, PBP4, and PBP2B in NMR buffer (10 mM MES pH 5.8, 25 mM NaCl) at 308 K with the peptides (SP1, SP2, SP4, SP5 and SP6 with PBP5, SP2 also with PBP4 and PBP2B) at 1:100 PBP5:peptide molar ratios using a pseudo 2D pulse sequence that leverages a shaped pulse train for saturation and selected water suppression by excitation sculpting with gradients. All STD experiments were performed at 800 MHz for best resolution and least signal overlap. Total saturation time for STD measurements was 2 s. A 2 s recycling delay, and 64 scans were used for all STD experiments. Selective protein saturation was achieved by setting the on-resonance

frequency on PBP5 methyl peaks at 0.73 ppm. For the off-resonance (reference) spectrum, the irradiation frequency was shifted to −40 ppm. The experimentally observed STD NMR effect is analyzed by converting it into the STD amplification factor ($STD_{amp}$) using Eq. 3[24]:

$$STD_{amp} = \frac{I_0 - I_{Sat}}{I_0} \times \frac{[L]_{Total}}{[P]_{Total}} \qquad (3)$$

where $I_0$ and $I_{Sat}$ are the ligand peak intensities in an off-resonance reference and on-resonance spectrum, respectively. $([L]_{Total}/[P]_{Total})$ is the ligand excess relative to a constant protein concentration.

For a uniform analysis, we normalized the STD amplification factor by using the highest STD amplification factor in the SP1:PBP5 analysis and using the same STD amplification factor to normalize other SPs (SP2-6) to allow for a rigorous comparison. For the analysis of the STD NMR data only data from non-exchangeable protons (CH groups) were utilized. Overlapped peaks were completely excluded from the analysis. Only protons with a STD that changed more than ≥50% are considered significantly changed for our analysis.

### Reporting summary
Further information on research design is available in the Nature Portfolio Reporting Summary linked to this article.

## Data availability
All NMR chemical shifts have been deposited in the BioMagResBank BMRB 51690 (sequence-specific $^1$H, $^{13}$C, and $^{15}$N backbone resonance assignments of 70 kDa Penicillin Binding Protein PBP5) and BMRB 51692; [https://doi.org/10.13018/BMR51692] (Sidechain Ile, Leu, and Val methyl chemical shift assignments for Penicillin Binding Protein PBP5). Atomic coordinates and structure factors have been deposited in the Protein Data Bank 6MKA and 6MKG. Source data are provided as a Source data file and on Figshare (https://doi.org/10.6084/m9.figshare.24926319). Source data are provided with this paper.

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

## Acknowledgements
We would like to thank Page and Peti Laboratory coworkers that helped with the expression and purification efforts for this project. This work is supported by grant RO1AI141522 and from the National Institute of Allergy and Infectious Diseases to WP and LBR.

## Author contributions
R.P., M.A., L.B.R. & W.P. developed the concept. Y.H. performed NMR data collection and analysis of the NMR data, including STD NMR data. M.F. synthesized the peptides. P.U.-S. and C.D. performed disk assays, MICs and WB analysis. G.S.K. performed NMR data collection of SP1, SP4, and SP5 titration with PBP5. M.T.K. performed protein expression for NMR spectroscopy and PG titration NMR experiments. M.S.C. expressed proteins for NMR data collection studies and performed assays. Y.L. extracted PG from *E. faecium*. W.P. and R.P. wrote the manuscript with comments and inputs from all co-authors.

## Competing interests
The authors declare no competing interests. The funding agencies had no role in study design, data collection and analysis, decision to publish, or preparation of the manuscript.
