## [Transparent Peer Review file · Nature Communications]

Peptidoglycan recruitment by a Penicillin Binding Protein

Corresponding Author: Dr Wolfgang Peti

Version 0:

Reviewer comments:

Reviewer #1

(Remarks to the Author)

The authors present a very interesting analysis of key peptidoglycan binding sites on penicillin-binding protein 5 (PBP5). Notably, they report the NMR spectral assignment of a 304-residue protein construct enabling characterization of the peptidoglycan binding interface, which had previously been inaccessible by other structural techniques due to its weak affinity. This study provides another potentially valuable avenue for therapeutic development targeting stem peptide binding sites to disrupt peptidoglycan synthesis and, in turn, the pathogenicity of ESKAPE pathogens. The manuscript is well written and very interesting. I recommend addressing a number of questions prior to publication..

1. A significant portion of the first paragraph refers to Figure 1A; however, in many instances, the figure does not fully illustrate the points. In particular, the full maturation process is not depicted, which limits the readability of this section. Including a more comprehensive figure would help readers better understand the assembly and regulation of peptidoglycan by PBPs.

2. The manuscript frequently suggests that PBP5's capacity to recruit peptidoglycan explains why peptidoglycan cross-linking is not affected by β -lactam resistance. For instance, on page 4, the authors write, "Thus, the PBP5 variants resistant to β -lactams are fully capable of PG recruitment, explaining in part why PG cross-linking is not negatively impacted by mutations responsible for the acquisition of β -lactam resistance."

I found this reasoning difficult to follow. As stated in the introduction and cited literature, PBP5 catalyzes cross-linking via its active-site Ser, and PBP5's β -lactam resistance stems from reduced affinity for β -lactams – not from changes to substrate binding or recruitment. If the resistance mutations do not impair cross-linking activity, then PBP5 should function normally regardless of where or how stem peptide recruitment occurs. In this context, the physical separation of the recruitment region, catalytic site, and resistance-conferring mutations does not obviously explain the preservation of activity.

Please clarify the intended connections here and in other similar statements linking PG recruitment and cross-linking activity in resistant variants. It would be helpful to clarify this in the introduction.

3. The introduction would benefit from a brief mention of the orthogonal strategy to block transpeptidation, as described in the first paragraph of the discussion. Introducing this concept earlier would help frame the study's rationale more clearly and provide a stronger justification for the experimental focus.

4. The authors demonstrate that PBP5 NMR spectra have been assigned. However, no spectral data, assignment tables, or references to BMRB deposition are provided. Please clarify whether the assignments have been deposited, and if not, please submit them.

5. Does the PBP5 used in this study retain catalytic activity? If so, is there any indication of cross-linking between stem peptides throughout the NMR experiments? Cross-linking could alter the size and diffusion of the stem peptides and their complexes, which could explain the intermediate exchange observed with the reduced PG. Clarifying whether cross-linking occurs under these conditions would help interpret the NMR results more accurately.

6. Figure 6C presents an intriguing conceptual model in which PBP5 acts analogously to a zipper, joining two SP chains. I recommend expanding on this model in the discussion. A more detailed mechanistic explanation would be highly informative and of interest to readers, especially those less familiar with transpeptidase activity or the engagement of PBP

substrates.

Instead of highlighting only selected key residues on the protein surface representation, it may be more informative to display the full mapped CSP data, color-coded by magnitude. This will likely facilitate the visualization and localization of the binding interface. This approach may also provide additional data to consider when modeling the mechanism of SP binding sites, as shown in Figure 6C.

On this note, while a distinct SP-binding site from the catalytic site is mentioned, a clear structural binding mechanism is never proposed. Based on Figure 6C, one might infer the presence of two opposing SP binding interfaces positioned perpendicular to the catalytic groove, thereby facilitating the transpeptidation of adjacent chain residues. If this is the authors' interpretation, I encourage them to elaborate on this model in greater detail and discuss its potential implications.

7. Are Glu284, Glu291, Ser294, and Asn295 supposed to be residues 384, 391, 394, and 395? If not, please provide data for them in Figure 1 or explain their omission.

8. What do the ratios represent? For example, the stem peptide titration was carried out up to a 1:80 ratio but it is unclear whether this represents a molar ratio or a weight-to-weight ratio.

9. Based on the Table S1 title, the reported K_d value (3.9mM) was determined from NMR CSPs. However, the statement that "For practical reasons (synthesis of peptides), most titrations were stopped at a ratio of 1:80 (PBP5:SP peptide), with most CSPs still not in full saturation, highlighting a weak interaction ($K_D \sim 3.9 \pm 1$ mM, Table S1)." Suggests that saturation was not reached for most residues. This significantly undermines the reliability of the fit. Without approaching saturation, any estimation is likely to yield an inaccurate and misleading affinity value. While the experimental limitations are understandable, it does not justify reporting a quantitative K_d that lacks sufficient supporting data. I suggest either omitting the value or presenting it as a rough estimation.

10. Regarding purified peptidoglycan-related data, the rationale for testing reduced and oxidized peptidoglycans from *E. faecium* is not clear. Please explain the motivation for this comparison and cite relevant literature.

11. Similarly, what does the ratio represent? If they represent molar ratios, how were the concentrations determined?

12. Increased spectral changes are reported with the reduced PG relative to the oxidized form. Based on my understanding, the reduction with borohydride produces muramitol forms not naturally present in peptidoglycans. Why would these modifications cause additional changes? What biological relevance does this have?

13. On page 14, the authors suggest that the reduced PG was used to probe the role of the disaccharide moiety. If so, it would seem more relevant to test pure polysaccharides instead. Could the authors comment on this?

14. Broadening of cross-peaks is attributed to intermediate exchange NMR time scale. Have the authors tested the binding at a higher or lower temperature to alter the exchange kinetics?

15. It would be helpful to provide a comparative chart of peak intensities across conditions to better illustrate the changes.

16. Regarding the oxidized PG dataset, in Figure 2B, several peaks appear to diminish in intensity, similar to those observed in the reduced PG sample. Can the authors confirm this?

17. The authors stated, "While limited, the CSP pattern of oxidized-PG results corresponded to a subset of residues that were also identified in the SP1 interaction (Figs. 1g,2f)." However, Figure 2F appears to highlight only two residues. This limited overlap may stem from the relatively lower stoichiometric ratio. I recommend increasing concentration of PG to strengthen this analysis and better support their statement.

18. The authors used several PBP5 variants for *E. faecium* viability assay. However, it is unclear how the authors decided on their designs. Could the authors explain the rationale behind the chosen mutations and how specific residue groupings were determined?

19. The STD NMR spectra do not clearly show any differences between the saturated and unsaturated conditions. Where is the $\geq 50\%$ STD difference as reported by the authors? Also, where are the control saturated and unsaturated STD NMR spectra (i.e., stem peptide only, not stem peptide+PBP)? These controls are necessary to validate the reported binding.

20. The HSQC spectra of PBP4 and PBP2B should be presented to better support the findings.

21. The authors state that PBP5 mediates the intrinsic β -lactam resistance in *E. faecium*. Could the authors clarify whether this resistance arise from an altered affinity for β -lactams or the gain of β -lactamase activity?

22. There are many instances where abbreviations are not defined. For example, the TP and N1 domains, as well as L1 and L2, are never defined.

23. There are also instances where defined abbreviations are written in their expanded forms instead of their abbreviations.

24. There are several instances of concluding sentences in the results section interpreting the results. These should be

reserved for the discussion section.

25. S295 should be S294 in Figure 3D.

Reviewer #2

(Remarks to the Author)

Review of: Peptidoglycan recruitment by a Penicillin Binding Protein

This study examines the molecular mechanisms by which Penicillin Binding Protein 5 (PBP5) from *Enterococcus faecium* recruits peptidoglycan (PG) and its relationship to antibiotic resistance mechanisms. PBP5, a low-affinity class B PBP, is responsible for intrinsic β -lactam resistance in *E. faecium*. The research shows that PG binds to PBP5 at the periphery of the active site rather than directly within it. Mutations in recruitment sites lead to failure in PG cell wall formation and bacterial death, while variants that disrupt these sites exhibit increased susceptibility to β -lactams. The research indicates that the D-Ala-D-Ala moiety is not critical for stem peptide recruitment. Instead, the main interaction occurs at the L-Lys(D-iAsn) residue, with experiments showing that peptides lacking D-Ala-D-Ala still bind effectively to PBP5. The study shows that stem peptides can bind PBP5 even in the presence of β -lactam antibiotics, as the recruitment mechanism operates through different interaction sites than β -lactam binding – this is an important finding for the field. The L-Lys(D-iAsn) residue proves essential for recruitment, with replacement by glycine significantly reducing PBP5 interaction, while the D-iGln residue contributes to binding but to a lesser extent. The research notes broad substrate specificity across PBPs from different bacterial species, with PBP4 from *E. faecalis* capable of cross-linking *E. faecium* stem peptides, indicating conserved recognition elements. This conservation may represent potential therapeutic targets, as the L-Lys residue appears to be a factor in cross-reactivity of PBPs from different organisms, potentially providing avenues for antibiotic development targeting conserved PBP recruitment sites as strategies to address antibiotic-resistant bacteria.

Major concerns

- Overgeneralization of their conclusions about a β -lactam-resistant PBP (PBP5) to all PBPs
 - o The authors draw a conclusion broadly about the mechanisms of PBP-substrate recruitment based on their data, which uses the one exception to PBPs that is actually resistant to β -lactam antibiotics (PBP5)
 - o If they wish to validate such a claim, the authors should use other types of PBPs which are not resistant to β -lactam antibiotics
 - o "...suggesting that the mechanism underlying the efficacy of PBP inactivation by β -lactam antibiotics is not exclusively rely on substrate binding mimicry." (pg. 11)
- In pages 6 and 7, the authors presented the chemical shift perturbation (CSP) resulting from the stem pentapeptide SP1 interacting with the enzyme. The authors state that the CSP of the SP1 and that of two antibiotics were similar, yet, concluded that stem peptides (overgeneralization) and beta lactams bind to distinct surfaces within the enzyme. The authors should further compare and contrast the CSPs to highlight exactly the point of similarities and differences. Furthermore, the author should limit that conclusion to SP1 as the natural peptide contains sugar moieties at the N-terminus.
- The chemical shift observed by SP1 should further be validated showing the lack of, or change in, chemical shift perturbation when a carefully designed negative control for SP1 is used. Furthermore, given that the author challenges the role of the catalytic serine residue in recruitment, perhaps a deleted mutant of that residue should be used to further show that.
- Figure 2: The authors use the NMR data from Figure 2 to state in pages 14 and 15 that "Consistent with this result, we determined that the stem peptide interaction with PBP5 is weak (~4 mM), with the recruitment of reduced PG being comparatively enhanced, suggesting that the disaccharide moiety contributes to PG precursor binding". We must challenge the validity of this conclusion given that such analysis seems to have been done using non-purified peptidoglycan fragments.
 - o Per the material and methods section, "to reduce the muramic acid into alcohols the isolated muropeptides was treated with 250mM sodium borate buffer and 10mg/mL for 30 minutes then were lyophilized". Though this method is well established as a means to reduce the sugar aldehyde and ketones into alcohol, it does not provide a pure material necessary for detection on subtle molecular interactions that the authors want to highlight.
 - o The isolation of the PG results in a heterogeneous mixture of glycopeptides (penta, tetra, tri, di, crosslinked or non-crosslinked) along with the cognate sugars. The authors did not comment on which one of these peptides was used to provide the data shown in Figure 2.
 - o Wall teichoic acids are conjugated onto the sugar of the peptidoglycan. These molecules are often removed from the PG through acid treatment which hydrolyzes them off the PG. The method section of the paper does not include such a step, leaving us wondering about the contribution of the teichoic acids on the chemical shift perturbations observed?
- The KD value for PBP5 and SP is reported, but this was without saturating substrate (pg. 6). The statement from the Methods that the values are overestimated should be included in the main text ("In our analysis specifically, the reported KD values are overestimated because our titrations even at the highest peptide to protein ratios (e.g., 1:80 for PBP5:SP1) did not reach saturation").
- Figure 3
 - o Given that PBP5 catalyzes the crosslinking of the PG upon beta lactam challenges, the authors use a disk diffusion susceptibility assay to demonstrate the viability of the bacteria as a result of the PG formation. However, this assay does not isolate the step of the recruitment of the stem peptides by residues separate from the catalytic residue, it simply points out PBP5's resistance to the beta lactams used (which is already known in the literature and not a molecular understanding of how this PBP recognizes PG fragments). As such, conclusions regarding this specific step of the transpeptidation cannot be drawn from this data. The language needs to reflect this lack of absolute mechanistic connection.
- Figure 4

- o For variant SP4, why was L-Lys(D-iAsn) replaced with Gly? Due to the lack of side chain resulting in extreme flexibility of Gly, this can result in unpredictable effects for functional interpretation. We instead suggest utilizing Ala in this position to preserve stereochemistry. The author may acetylate the amine of the iAsn moiety to isolate the role of the amine on the iAsn. Other strategies include using a peptide with different stereochemistry to clearly show the role of these residues.
- o A change of D-iGln to D-nGln resulted in a change in CSP highlighting the participation of this residue in the recruitment. Can the authors consider using a better control such as L-iGln or a more drastic change in the charge state of that amino acid by using D-isoglutamic acid or D-glutamic acid to strengthen that conclusion.
- o The authors used SP2 to show that the terminal D-Ala-D-Ala is not necessary for recruitment. In general, the authors fail to clarify whether these peptides are recruited as acyl donor or acceptors. Does the pentapeptide and the tripeptide function as both donor and acceptor? Prior literature has shown that fluorescently labeled probes without the terminal D-ala-D-ala are able to label the PG. Are the authors challenging or substantiating these claims by NMR?

Minor concerns

- We think the authors should improve the clarity of the manuscript by correcting grammar mistakes and improving the clarity of their conclusions to help the readers understand the story. For instance, "...suggesting that the mechanism underlying the efficacy of PBP inactivation by β -lactam antibiotics is not exclusively rely on substrate binding mimicry." (pg. 11)
- The authors refer to SP as "stem pentapeptides" while some of the peptides used here are not pentapeptides (SP2 and SP3). Please clarify these abbreviations
- Figure 2a
 - o We believe the authors should further discuss the necessity for the reduction of the isolated stem peptide in the text. If the reduced PG is used for technical/experimental purposes for the NMR, this needs to be explained.
 - o What is "bh" above the arrow in the figure? If referring to sodium borohydride, please use the chemical formula (NaBH_4).
- Figure 3
 - o We suggest placing labels on the gel rather than in the figure caption in order to increase the clarity
- Figure 4
 - o What is "nGln" in variant SP6? Please include the structure of this
 - o Why are variants SP4 and SP5 included in Figure 4 when it isn't mentioned until Figure 5?
- Figure 5e
 - o The authors include a label for "L-Gly". However, since Gly has no stereocenter, this should simply be "Gly".
 - On pg. 12, the figure references for the following sentence are incorrect. It should refer to the supplemental figure S10, as there is no Fig. 10 in the main text: "Using 1D 1H STD NMR experiments we showed that SP6 has an interaction pattern similar to that of SP1 and includes protons from C β in D-iGln (Figs. S10f,g).

Reviewer #3

(Remarks to the Author)

I have a simple question: does the PBP5 preparation exhibit any enzymatic activity (transpeptidation and/or hydrolysis) on peptide SP1? If it does, it might be significant at the large PBP5 concentrations used. By the way, how long did the recordings of the NMR spectra take?

Reviewer #4

(Remarks to the Author)

Version 1:

Reviewer comments:

Reviewer #1

(Remarks to the Author)

I appreciate the detailed response to my comments, along with the additional data, clarifications, and manuscript revisions. Several of my comments were addressed, including inclusion of BMRB accession numbers and source data. Other important questions remain unanswered and are outlined below.

(1) The figures illustrating 1H-15N CSP plots do not appear to align with the source data. For instance, in Figure 1E, the annotated bars with residue identities do not show significant differences when compared to the source data. Moreover, the HSQC spectra, with chemical shifts from BMRB entry 51690, also reveal little to no changes for these residues. These discrepancies need to be explained.

(2) For some residues (eg 523N) it is difficult to discern the spectral changes due to significant overlap with other signals. How were CSPs measured for overlapping peaks? Providing separate spectra for each titration point may illustrate this point.

(3) Are the data from the ILV 13C-1H spectra plotted correctly? They seem to be presented as a bar graph rather than as an XY plot, and the residue numbers are not continuous. In figure 1F, the numbers jump from 367 to 427 (an increase of 60), then to 449 (an increase of 22). This is confusing and the plots do not appear to accurately map CSP magnitudes to residue number. Also, the source data lists two values for the same residue in several cases. How are these depicted in the figure?

(4) Based on BMRB entry 51690 and the source data, residues G98 and G268 undergo a substantial shift in response to SP, showing that some of the biggest CSP values in the dataset occur outside the 349-673 region displayed in the manuscript. These changes are potentially important and should be discussed, but they are not mentioned in the manuscript.

(5) My previous question (#12) was not addressed – why does reduced PG (muramitol form) produce greater spectral changes than the oxidized form? Do additional CSPs reflect a structural property of the reduced sugar, a difference in stability/solubility, or another factor?

(6) My previous comment (#15) was not addressed – suggestion to include a chart of peak intensities for each amino acid. to provide an additional layer of information regarding binding. I believe this would help illustrate the effects more clearly.

(7) Thank you for the response to my comment (#18). However, I am still unclear about the rationale for the specific design of the PBP5 variants used in the viability assay. My original question was more related to how the authors decided on the six particular constructs tested, especially given that some variants contain a single mutation while others include two or more. For example, N523A alone had no effect, whereas N523A/Q627A did. However, the Q627A single mutant was not tested. Please clarify how these specific combinations were chosen, and why certain potentially informative single mutants (such as Q627A) were not examined.

Reviewer #2

(Remarks to the Author)

My concerns have been addressed.

Reviewer #4

(Remarks to the Author)

Reviewer #1 (Remarks to the Author):

The authors present a very interesting analysis of key peptidoglycan binding sites on penicillin-binding protein 5 (PBP5). Notably, they report the NMR spectral assignment of a 304-residue protein construct enabling characterization of the peptidoglycan binding interface, which had previously been inaccessible by other structural techniques due to its weak affinity. This study provides another potentially valuable avenue for therapeutic development targeting stem peptide binding sites to disrupt peptidoglycan synthesis and, in turn, the pathogenicity of ESKAPE pathogens. The manuscript is well written and very interesting. I recommend addressing a number of questions prior to publication..

We thank the reviewer for its support of our data and our careful analysis and appreciate the strong support of our work. We want to highlight that PBP5 studied in this manuscript is 645 amino acids; studying a protein of this size using NMR spectroscopy is even more complicated than that of a 304-amino acid protein.

We thank the reviewer for highlighting that the manuscript is 'well written' and 'very interesting'.

1. A significant portion of the first paragraph refers to Figure 1A; however, in many instances, the figure does not fully illustrate the points. In particular, the full maturation process is not depicted, which limits the readability of this section. Including a more comprehensive figure would help readers better understand the assembly and regulation of peptidoglycan by PBPs.

We thank the reviewer for highlighting this issue. As suggested, we have reduced the number of times we call out Figure 1a. As the full maturation process of PG has been summarized in many reviews (that are cited), we made sure that appropriate citations are in place.

2. The manuscript frequently suggests that PBP5's capacity to recruit peptidoglycan explains why peptidoglycan cross-linking is not affected by β -lactam resistance. For instance, on page 4, the authors write, "Thus, the PBP5 variants resistant to β -lactams are fully capable of PG recruitment, explaining in part why PG cross-linking is not negatively impacted by mutations responsible for the acquisition of β -lactam resistance."

We thank the reviewer for highlighting this important finding in our data/manuscript. We understand that this sentence taken out of context and read by itself might not make much sense. But it does make sense when combined with the sentence before it, which reads as follows: "We discovered that the PBP5 residues that mediate PG/SP recruitment, a step that precedes cross-linking, are distal from the nucleophilic Ser422 and distinct from most mutations that are responsible for increased β -lactam resistance." Read together, these sentences highlight that PG binding outside of the active site is critical for bacterial survival, as our data shows.

I found this reasoning difficult to follow. As stated in the introduction and cited literature, PBP5 catalyzes cross-linking via its active-site Ser, and PBP5's β -lactam resistance stems from reduced affinity for β -lactams – not from changes to substrate binding or recruitment. If the resistance mutations do not impair cross-linking activity, then PBP5 should function normally regardless of where or how stem peptide recruitment occurs. In this context, the physical separation of the recruitment region, catalytic site, and resistance-conferring mutations does not obviously explain the preservation of activity.

The reviewer is correct – PBP5 catalyzes stem peptide cross-linking, which requires the active site Ser residue. The reviewer is also correct that resistant PBPs have altered interactions with β -lactams without significantly impacting endogenous substrate crosslinking (which is why the bacteria grow).

However, critically what the reviewer might not have fully considered is that β -lactams exclusively bind PBPs using a covalent bond to the active site Ser residue (i.e. β -lactams do not bind to PBPs in which the active site Ser is mutated). So, changes to the active site – should lead to a reduction of cross-linking and thus automatically to a less viable bacterium – however, resistant bacteria are viable, showing that a less effective cross-linking step in resistant bacteria is ‘good-enough’ for bacterial survival and not the rate limiting event. As our data show – see Figure 3 – PG/SP binding indeed is the rate limiting event, as when the binding sites are deleted the bacterium is dead.

Lastly, that the active site and the PG/SP recruitment/binding sites are distinct and complementary is directly shown in our NMR experiment. Here we show that SP peptides can be recruited to PBP5 in the presence of saturating conditions of β -lactams (Pen-G and ceftaroline), further highlighting the independence of binding sites. We hope that this clarification is helpful.

Please clarify the intended connections here and in other similar statements linking PG recruitment and cross-linking activity in resistant variants. It would be helpful to clarify this in the introduction.

We have added additional guidance throughout the manuscript.

3. The introduction would benefit from a brief mention of the orthogonal strategy to block transpeptidation, as described in the first paragraph of the discussion. Introducing this concept earlier would help frame the study’s rationale more clearly and provide a stronger justification for the experimental focus.

We agree with the reviewer that a sentence that highlights this issue in the introduction is important and reads now as follows.

`'Thus, defining the molecular basis of the recognition of the cross-linking of PG/SP is essential for developing novel antibiotics that exhibit high barriers to resistance.'`

4. The authors demonstrate that PBP5 NMR spectra have been assigned. However, no spectral data, assignment tables, or references to BMRB deposition are provided. Please clarify whether the assignments have been deposited, and if not, please submit them.

All of our data are fully available in public databases and repositories, with the sequence specific backbone assignment and the ILV assignment having been deposited in the BMRB. As requested, we have added the BMRBID to the manuscript.

The sequence-specific backbone assignment of PBP5 was previously published and is publicly available BMRBID:51690.

5. Does the PBP5 used in this study retain catalytic activity? If so, is there any indication of cross-linking between stem peptides throughout the NMR experiments? Cross-linking

could alter the size and diffusion of the stem peptides and their complexes, which could explain the intermediate exchange observed with the reduced PG. Clarifying whether cross-linking occurs under these conditions would help interpret the NMR results more accurately.

As repeatedly described in the field – free PBPs, such as PBP5, do not have any significant activity that can be readily observed in vitro. To ensure that this is also the case for *E. faecium* PBP5 used in this study, we performed cross-linking experiments and tested the formation of peptidoglycan cross-links by high resolution mass spectrometry, as previously described (<https://doi.org/10.1074/jbc.M507384200>). As anticipated, no cross-linked products were identified. The reason why these enzymes are so inefficient in vitro is not understood, but this result is completely consistent with literature reports for the last 20-25 years.

36. Figure 6C presents an intriguing conceptual model in which PBP5 acts analogously to a zipper, joining two SP chains. I recommend expanding on this model in the discussion. A more detailed mechanistic explanation would be highly informative and of interest to readers, especially those less familiar with transpeptidase activity or the engagement of PBP substrates. Instead of highlighting only selected key residues on the protein surface representation, it may be more informative to display the full mapped CSP data, color-coded by magnitude. This will likely facilitate the visualization and localization of the binding interface. This approach may also provide additional data to consider when modeling the mechanism of SP binding sites, as shown in Figure 6C.

Detailed interaction surfaces are presented in Figures 1, 2, 4 and 5 as well as in Supplemental Figures S2 and S10. Figures 6c/d are conceptual figures to summarize the key results from the work, but the molecular and mechanistic data is in the data figures. Regarding the full CSP mapping – please see Supplemental Figure S2; this figure shows the observed changes in the interaction surface upon titration with increasing amounts of PG.

Furthermore, all raw data is available on Figshare (as cited in the manuscript) and thus any reader can use the available data to create any desired figure.

We hope that this summary addresses the concern of the reviewer; we agree with the reviewer that it is never easy to convey such a large amount of data and we all have our preferred views – we hope that we found enough common ground in the manuscript.

On this note, while a distinct SP-binding site from the catalytic site is mentioned, a clear structural binding mechanism is never proposed. Based on Figure 6C, one might infer the presence of two opposing SP binding interfaces positioned perpendicular to the catalytic groove, thereby facilitating the transpeptidation of adjacent chain residues. If this is the authors' interpretation, I encourage them to elaborate on this model in greater detail and discuss its potential implications.

Figure 6C is a simplified overview model that conveys the conclusions based on our experimental data to a broader audience. Our NMR-based titration studies define residues on the PBP5 surface that are critical for the recruitment of PG/SP. Indeed, the key point here is the data shown in Figure 3 – when these experimentally defined residues are mutated – *E. faecium* is not viable as it is unable to recruit PG. This directly correlates our NMR studies with physiological outcomes in *E. faecium*. At this point we did not perform any kind of NMR-based docking experiments, as it is established these methods do not effectively capture dynamic interactions like those studied

herein (i.e., a peptide binding to a protein). Rather we feel that the work of confirming the NMR data in the biological organism, *E. faecium*, provides this biological and mechanistic information.

7. Are Glu284, Glu291, Ser294, and Asn295 supposed to be residues 384, 391, 394, and 395? If not, please provide data for them in Figure 1 or explain their omission.

We thank the reviewer for the question. Glu284, Glu291, Ser294, and Asn295 are PBP5 residues Glu284, Glu291, Ser294, and Asn295. We would like to draw attention to Figure 3, which highlights the residues and further enables the reader to fully understand our data.

8. What do the ratios represent? For example, the stem peptide titration was carried out up to a 1:80 ratio but it is unclear whether this represents a molar ratio or a weight-to-weight ratio.

We thank the reviewer for her/his question, and we now clarify that this information refers to molar ratios.

9. Based on the Table S1 title, the reported K_d value (3.9mM) was determined from NMR CSPs. However, the statement that “For practical reasons (synthesis of peptides), most titrations were stopped at a ratio of 1:80 (PBP5:SP peptide), with most CSPs still not in full saturation, highlighting a weak interaction (K_D ~ 3.9 ± 1 mM, Table S1).” Suggests that saturation was not reached for most residues. This significantly undermines the reliability of the fit. Without approaching saturation, any estimation is likely to yield an inaccurate and misleading affinity value. While the experimental limitations are understandable, it does not justify reporting a quantitative K_d that lacks sufficient supporting data. I suggest either omitting the value or presenting it as a rough estimation.

The reviewer is correct that it is always important to try, under all circumstances, to achieve binding saturation. When possible, a fast timescale NMR titration, such as the one between PBP5 and the stem peptide, is performed at increasing titrant concentrations until saturation is achieved. However, sometimes saturation cannot be achieved due to technical issues, including but not limited to spectral sensitivity, availability of substrate, substrate solubility, etc. Indeed, while full saturation was not achieved in our analysis, the only consequence of not achieving full saturation is that the reported K_D will be higher (weaker) than the actual K_D, as the fit is limited due to a lack of a saturation point. That is, this data provides the upper bound on the K_D. So indeed, it is apparent that this number is an overestimation, and the ‘real’ K_D is lower than the one reported.

10. Regarding purified peptidoglycan-related data, the rationale for testing reduced and oxidized peptidoglycans from *E. faecium* is not clear. Please explain the motivation for this comparison and cite relevant literature.

The *in vivo* cross-linking activity of PBPs might also depend upon the interaction of the enzyme with sugar components of the substrates, namely the glycan strands in nascent peptidoglycan. In these substrates, the disaccharide-peptide subunits are connected by β1,4 glycosidic bonds. Testing both reduced and oxidized peptidoglycan fragments offered the possibility to test two mimics of the natural substrates instead of one.

11. Similarly, what does the ratio represent? If they represent molar ratios, how were the concentrations determined?

We thank the reviewer for her/his question. All ratios are molar ratios and it has now been defined multiple times in the manuscript to ensure this is clear.

12. Increased spectral changes are reported with the reduced PG relative to the oxidized form. Based on my understanding, the reduction with borohydride produces muramitol forms not naturally present in peptidoglycans. Why would these modifications cause additional changes? What biological relevance does this have?

We thank the reviewer for highlighting that it is important to consider the biological relevance for the experimental design. Importantly, and as also stated in our answer to comment #10, the natural substrates are glycan strands containing disaccharide subunits connected by β 1,4 glycosidic bonds. Thus, the presence of neither muramitol nor MurNAc at the reducing end of the disaccharide peptides, as used in our experiments, corresponds “to the forms naturally present in peptidoglycan”. This is the rationale for testing both and reporting the corresponding data and thus our data has full biological relevance.

13. On page 14, the authors suggest that the reduced PG was used to probe the role of the disaccharide moiety. If so, it would seem more relevant to test pure polysaccharides instead. Could the authors comment on this?

We thank the reviewer for this suggestion – however, in our studies, we test the role of the disaccharide moiety for binding in context of the stem peptide. Thus, testing disaccharide binding without the peptide moiety will not address the biological question we are asking. Of note, testing the combination of peptides and disaccharides as separate molecules is unlikely to closely mimic the interaction with the natural substrate, in particular because of the presence of a carboxyl group on the lactoyl moiety of MurNAc in free disaccharides.

14. Broadening of cross-peaks is attributed to intermediate exchange NMR time scale. Have the authors tested the binding at a higher or lower temperature to alter the exchange kinetics?

We thank the reviewer for this suggestion. While it is readily possible to change the temperature of the NMR measurements for smaller proteins (100-200 aa), for ~70+ kDa proteins, such as PBP5, reducing the temperature by 10°C will double the overall tumbling time – so the protein will behave like a 150 kDa protein, with much deteriorated spectral qualities. Increasing the temperature of the measurement was not possible, as we do want to be at least 25°C lower than the unfolding temperature of PBP5 to ensure that the majority (95%) of the protein is in a folded and substrate binding competent conformation. Furthermore, each measurement time point was ~12-20 hrs of 800 MHz NMR time. Taking together our data was recorded in the most optimal manner possible—ensuring protein stability and allowing for a spectral quality that ensures rigorous data analysis—to understand binding. As changing the temperature was not experimentally feasible, we leveraged an alternative approach by testing a large array of different ligands to present a complete, molecular understanding of which elements of the SPs are critical for PBP binding.

15. It would be helpful to provide a comparative chart of peak intensities across conditions to better illustrate the changes.

We are not certain that we understand the request of the reviewer and we hope to answer the request appropriately. We are not certain which conditions the reviewer is referring to. We have tried to illustrate all interactions as clearly as possible. However, this has limitations working with

a 645 aa protein versus much smaller proteins that are more commonly used in NMR based studies. The data was collected and is presented in manners that are well-established for NMR-based titration experiments.

16. Regarding the oxidized PG dataset, in Figure 2B, several peaks appear to diminish in intensity, similar to those observed in the reduced PG sample. Can the authors confirm this?

Yes, we can confirm the statement of the reviewer.

17. The authors stated, “While limited, the CSP pattern of oxidized-PG results corresponded to a subset of residues that were also identified in the SP1 interaction (Figs. 1g,2f).” However, Figure 2F appears to highlight only two residues. This limited overlap may stem from the relatively lower stoichiometric ratio. I recommend increasing concentration of PG to strengthen this analysis and better support their statement.

We have performed the measurement in the most rigorous manner. As seen in Figure S2, we have performed the interaction at a large range of concentrations. Extracting natural PG from *E. faecium* in these amounts has already taken more than a month of work and the experiments are 100s+ hours of NMR time. We can assure the reviewer that these experiments are performed as rigorously and carefully as possible.

18. The authors used several PBP5 variants for E. faecium viability assay. However, it is unclear how the authors decided on their designs. Could the authors explain the rationale behind the chosen mutations and how specific residue groupings were determined?

We thank the reviewer for her/his question. PBP5 variants selected correspond to the residues whose peaks exhibit the most significant changes in our NMR titration studies.

19. The STD NMR spectra do not clearly show any differences between the saturated and unsaturated conditions. Where is the $\geq 50\%$ STD difference as reported by the authors? Also, where are the control saturated and unsaturated STD NMR spectra (i.e., stem peptide only, not stem peptide+PBP)? These controls are necessary to validate the reported binding.

We thank the reviewer for her/his insightful comment and appreciate the opportunity to clarify our STD-NMR experimental design and data interpretation. We understand that visual differences in the STD spectra may not be immediately obvious. However, our intention was to illustrate that the broad protein signal (the “hump”) is effectively saturated when we apply saturation at 0.7 ppm. To emphasize this, we increased the overall intensity in the displayed spectra, which also led to an increase in the intensity of other NMR signals. This was necessary because we rigorously quantified the STD effect using integrals rather than relying solely on visual inspection. The integral values used for quantitative analysis are provided in the source files. Therefore, this figure demonstrates that saturation at 0.7 ppm successfully attenuates the protein signal, confirming effective excitation.

We appreciate the reviewer’s suggestion to check a control STD experiment with ligand alone. However, our ligand does not exhibit any ^1H NMR signals below 1.3 ppm. Since STD NMR (please see Mayer & Meyer, *Angew. Chem. Int. Ed.*, 1999, 38(12):1784-1788) relies on selective saturation transfer via spin diffusion, and the saturation frequency (0.7 ppm) is well-separated from any ligand signals, it is not possible for direct saturation to occur in the absence of protein.

Therefore, under these conditions, STD effects cannot arise unless mediated by protein–ligand binding. For this reason, we did not perform an STD experiment on the ligand alone, as no STD signal is expected.

20. The HSQC spectra of PBP4 and PBP2B should be presented to better support the findings.

The studies related to PBP2B and PBP4 that are presented in the manuscript did not require isotopic labeling, which is why the HSQC spectra of these proteins were not presented (i.e., neither protein was (²H, ¹⁵N)-labeled). We aim to perform such experiments in future studies.

21. The authors state that PBP5 mediates the intrinsic β -lactam resistance in *E. faecium*. Could the authors clarify whether this resistance arise from an altered affinity for β -lactams or the gain of β -lactamase activity?

There is no evidence, in >30 years of study, that *E. faecium* resistance is due to the gain of β -lactamase activity. Rather, it has been long established that PBP5 is the sole PBP required for *E. faecium* growth in the presence of cephalosporins, i.e. ceftriaxone, which have also been shown to inhibit all other *E. faecium* PBPs. Consequently, *E. faecium* strains from which PBP5 has been deleted are fully susceptible to ceftriaxone. As we show in Figure S3 – ceftriaxone does not bind PBP5. We hope this further clarifies our results.

22. There are many instances where abbreviations are not defined. For example, the TP and N1 domains, as well as L1 and L2, are never defined.

Thank you for the request – we now defined all domains and loops. Furthermore, N1 is now also annotated in Figure 1.

transpeptidase (TP)

PBP5 N1 domain (the N1 domain is one of 4 domains in PBPs and links the membrane-inserting helix with the TP domain)

...‘mediated by rigid surface loops in the PBP5 TP domain, L1 and L2, that facilitate communication between the TP and the N1 domain’

23. There are also instances where defined abbreviations are written in their expanded forms instead of their abbreviations.

Yes, the reviewer is correct, we sometimes refer to the full name to make it easier to remember, simply to improve overall manuscript readability. This is not uncommon, and we feel important for a broad journal such as Nature Communications, which is not only read by experts. We have certainly limited it as much as possible and revisited this issue in this revision again to further limit it as much as possible. Thank you.

24. There are several instances of concluding sentences in the results section interpreting the results. These should be reserved for the discussion section.

We completely agree with the reviewer that the results and discussion sections are separate entities, but a brief summary (i.e. a concluding sentence) of results, to facilitate readability, is very

common in many manuscripts. Thus, we have elected to keep these conclusion sentences in place as the summarize the results in a manner that facilitates readability for the broad audience of Nature Communications. Importantly, the summary sentences do not discuss the results in the framework of the field, which is only done in the discussion. Thank you.

25. S295 should be S294 in Figure 3D.

Updated – thank you!

Reviewer #2 (Remarks to the Author):

Review of: Peptidoglycan recruitment by a Penicillin Binding Protein

This study examines the molecular mechanisms by which Penicillin Binding Protein 5 (PBP5) from *Enterococcus faecium* recruits peptidoglycan (PG) and its relationship to antibiotic resistance mechanisms. PBP5, a low-affinity class B PBP, is responsible for intrinsic β -lactam resistance in *E. faecium*. The research shows that PG binds to PBP5 at the periphery of the active site rather than directly within it. Mutations in recruitment sites lead to failure in PG cell wall formation and bacterial death, while variants that disrupt these sites exhibit increased susceptibility to β -lactams. The research indicates that the D-Ala-D-Ala moiety is not critical for stem peptide recruitment. Instead, the main interaction occurs at the L-Lys(D-iAsn) residue, with experiments showing that peptides lacking D-Ala-D-Ala still bind effectively to PBP5. The study show that stem peptides can bind PBP5 even in the presence of β -lactam antibiotics, as the recruitment mechanism operates through different interaction sites than β -lactam binding – this is an important finding for the field. The L-Lys(D-iAsn) residue proves essential for recruitment, with replacement by glycine significantly reducing PBP5 interaction, while the D-iGln residue contributes to binding but to a lesser extent. The research notes broad substrate specificity across PBPs from different bacterial species, with PBP4 from *E. faecalis* capable of cross-linking *E. faecium* stem peptides, indicating conserved recognition elements. This conservation may represent potential therapeutic targets, as the L-Lys residue appears to be a factor in cross-reactivity of PBPs from different organisms, potentially providing avenues for antibiotic development targeting conserved PBP recruitment sites as strategies to address antibiotic-resistant bacteria.

We thank the reviewer for the careful review of our work, the nice summary and the strong support by highlighting that we present ‘important findings for the field’.

Major concerns

- **Overgeneralization of their conclusions about a β -lactam-resistant PBP (PBP5) to all PBPs**

We never intend to overgeneralize our findings – indeed even our title is referring to a single PBP (no plural used). We carefully describe using PBP5 of *E. faecium* throughout the manuscript.

- o **The authors draw a conclusion broadly about the mechanisms of PBP-substrate recruitment based on their data, which uses the one exception to PBPs that is actually resistant to β -lactam antibiotics (PBP5)**
- o **If they wish to validate such a claim, the authors should use other types of PBPs which are not resistant to β -lactam antibiotics**

We would like to clarify this comment from the reviewer, which is likely due to a misunderstanding. *Enterococcus faecium*, a member of the ESKAPE pathogens, are common causes of serious and difficult-to treat infections in modern hospitals and thus the decision was made to study the recruitment of a clinically relevant PBP. During the antibiotic era, *E. faecium* have developed high levels of resistance to β -lactam antibiotics through mutations of penicillin-binding protein 5 (PBP5), a transpeptidase that is able to function when β -lactam antibiotics effectively inhibit all other PBPs. Thus, PBP5 is indeed the only meaningful PBP to be studied in *E. faecium*.

The overarching goal of our work is to determine how PBP5 of *E. faecium* is both simultaneously resistant to β -lactam antibiotics yet still perform its critical function of cell wall synthesis. This question arises as β -lactams covalently bind to the active site of PBPs, thereby preventing cell wall synthesis. The field assumed that cell wall precursors (stem peptides/ PG fragments) are also recruited to PBPs by binding to the active site, as is typical for enzymes:substrates. Our key finding is that the precursors are recruited to a distinct/separate site on the PBP, directly explaining how the PBPs can be resistant to β -lactams yet still synthesize cell wall. Finally, we confirmed these findings physiologically by using alanine mutagenesis to alter structurally identified residues on PBP5 and demonstrating that these mutants were non-viable.

Thus indeed, when studying these effects in *Enterococcus faecium*, PBP5 is the only biologically viable candidate to be used for our studies.

o “...suggesting that the mechanism underlying the efficacy of PBP inactivation by β -lactam antibiotics is not exclusively rely on substrate binding mimicry.” (pg. 11)

The reviewer is correct that we show this by NMR spectroscopy – and confirm it by a variety of data points, including where we show SP binding to PBPs when the active site is occupied by a β -lactam.

• In pages 6 and 7, the authors presented the chemical shift perturbation (CSP) resulting from the stem pentapeptide SP1 interacting with the enzyme. The authors state that the CSP of the SP1 and that of two antibiotics were similar, yet, concluded that stem peptides (overgeneralization) and beta lactams bind to distinct surfaces within the enzyme. The authors should further compare and contrast the CSPs to highlight exactly the point of similarities and differences. Furthermore, the author should limit that conclusion to SP1 as the natural peptide contains sugar moieties at the N-terminus.

We thank the reviewer for her/his input, but we are worried that she/he might have misinterpreted our data. We report on a distinct set of CSPs for β -lactams (previously published) and for stem peptides (new data reported here). We highlight that a small number of CSPs are identified in both interactions, which are identified in areas distal to both the β -lactam and the PG/SP binding sites. We have studied these CSPs in detail (see citation 11) which are due to allosteric effects in the PBP5 N1 domain.

Indeed, the next sentence on page 7 states – “Because the vast majority of PBP5 resistance mutations are close to the PBP active site (10), our data show that β -lactam antibiotics and stem peptides bind distinct surfaces of PBP5, explaining, in part, why resistant variants still effectively process peptide cross-linking.” We hope this further clarifies these data.

• The chemical shift observed by SP1 should further be validated showing the lack of, or change in, chemical shift perturbation when a carefully designed negative control for SP1 is used. Furthermore, given that the author challenges the role of the catalytic serine

residue in recruitment, perhaps a deleted mutant of that residue should be used to further show that.

We fully agree with the reviewer that negative controls are essential for rigorous data, and that is exactly what has been done in our presented data. *First*, we repeated the data collection with PG extracted from natural source to ensure SP1 and PG have similar interaction patterns. *Second*, we confirmed these data by creating binding deletion variants in cells (in *E. faecium*) – see Figure 3. This effort directly correlated our in vitro NMR data into cellular outcomes. *Third*, we varied SP1 by creating 5 additional SP variants (SP2-6) by deleting/changing/modifying SP1 as negative/positive control experiments. These rigorous, orthogonal experiments are internally consistent, resulting in our confidence in our conclusions.

• Figure 2: The authors use the NMR data from Figure 2 to state in pages 14 and 15 that “Consistent with this result, we determined that the stem peptide interaction with PBP5 is weak (~4 mM), with the recruitment of reduced PG being comparatively enhanced, suggesting that the disaccharide moiety contributes to PG precursor binding”. We must challenge the validity of this conclusion given that such analysis seems to have been done using non-purified peptidoglycan fragments.

We appreciate the reviewers’ input, but we are not certain what the reviewer is specifically challenging here. We highlight that we leverage the SP1 CSP titration to estimate a K_D . As expected, the K_D is very low, exactly what is expected for the observed NMR data. As also highlighted in the response to Reviewer 1, in an optimal case we want to achieve binding saturation. Yet sometimes saturation cannot be achieved due to a variety of issues, including but not limited to spectral sensitivity, availability of substrate, substrate solubility, etc. However, the only effect of not achieving full saturation is that the calculated K_D will be larger than the experimental one, as the fit is limited due to a lack of saturation point. That is, this data provides an upper bound on the K_D . So indeed, it is completely clear that this number is an overestimation, and the ‘real’ K_D is lower (stronger) than the one reported.

Regarding the PG:PBP5 titration – we report how the PG was extracted and purified in the methods, which itself has been previously published and indeed PG extracted and purified in this manner has been used in different studies that are all published. Secondly, due to the change in the observed NMR based interaction pattern (change of molar ratios to achieve interactions and change in patterns of interaction), the K_D is clearly tighter; no other conclusion can be drawn. We did not attempt to quantify the new K_D as this would be an overinterpretation of the data – exactly in line with the request of the reviewer.

Lastly, if the reviewer is implying that ‘unknown molecules’ in the PG preparations bind to PBP5 and allosterically alter the binding of disaccharide-peptides, it would be necessary to provide a rationale for this concern. To the best of our knowledge, such an allosteric interaction with unknown components has never been described. We want to point out that our conclusion was exceedingly cautiously formulated “... suggesting that...”. As the reviewer is likely aware, it would be extremely challenging for any laboratory in the world to purify a specific disaccharide-peptide in sufficient amount for NMR analysis, which would likely take 24-36 months and thus has never been attempted.

In response to the reviewer, we have now edited this sentence as follows:

Consistent with this result, we determined that the SP interaction with PBP5 is weak (~4 mM), with the recruitment of reduced PG being

comparatively enhanced, suggesting that the additional elements present in PG, such as the disaccharide moiety, contribute to PG precursor binding.

o Per the material and methods section, “to reduce the muramic acid into alcohols the isolated muropeptides was treated with 250mM sodium borate buffer and 10mg/mL for 30 minutes then were lyophilized”. Though this method is well established as a means to reduce the sugar aldehyde and ketones into alcohol, it does not provide a pure material necessary for detection on subtle molecular interactions that the authors want to highlight.

We thank the reviewer for her/his concern, but we have compared the structure of reduced and unreduced disaccharide-peptides by mass spectrometry (MS). Reduction specifically leads to the reduction of MurNAc to muramitol.

o The isolation of the PG results in a heterogenous mixture of glycopeptides (penta, tetra, tri, di, crosslinked or non-crosslinked) along with the cognate sugars. The authors did not comment on which one of these peptides was used to provide the data shown in Figure 2.

The reviewer is correct in stating that we have not individually tested peptidoglycan fragments with various polymerization status (monomer *versus* dimer) or maturation of stem peptides (tripeptide, tetrapeptide *versus* pentapeptide). Indeed, the results obtained using our carefully prepared synthetic peptides indicate that these various species bind to PBP5 because the main interaction occurs at the L-Lys(D-iAsn) residue.

o Wall teichoic acids are conjugated onto the sugar of the peptidoglycan. These molecules are often removed from the PG through acid treatment which hydrolyzes them off the PG. The method section of the paper does not include such a step, leaving us wondering about the contribution of the teichoic acids on the chemical shift perturbations observed?

Our study focuses on the interaction of PBP5 with the peptide moiety of the substrate. Testing whether modification of the sugar moiety, *i.e.* acetylation, deacetylation, conjugation to lipoteichoic acid, modulates the affinity of PBP5 for peptidoglycan components is clearly an interesting question, but at this point also beyond the scope of the current study.

• The K_D value for PBP5 and SP is reported, but this was without saturating substrate (pg. 6). The statement from the Methods that the values are overestimated should be included in the main text (“In our analysis specifically, the reported K_D values are overestimated because our titrations even at the highest peptide to protein ratios (e.g., 1:80 for PBP5:SP1) did not reach saturation”).

Again, as we state, the K_D is an overestimation, *i.e.* the real K_D will be lower. We are willing to remove the K_D information, as it does not impact the biological outcome of our work, but we included it in the resubmission for completeness of the analysis, including highlighting the fact that it will always be better/tighter in reality than the number we report. Simply, our measurement provides an upper bound.

• Figure 3

o Given that PBP5 catalyzes the crosslinking of the PG upon beta lactam challenges, the

authors use a disk diffusion susceptibility assay to demonstrate the viability of the bacteria as a result of the PG formation. However, this assay does not isolate the step of the recruitment of the stem peptides by residues separate from the catalytic residue, it simply points out PBP5's resistance to the beta lactams used (which is already known in the literature and not a molecular understanding of how this PBP recognizes PG fragments). As such, conclusions regarding this specific step of the transpeptidation cannot be drawn from this data. The language needs to reflect this lack of absolute mechanistic connection.

We appreciate the concern of the reviewer, but we disagree with the assessment of the reviewer. We state in the manuscript, 'While PBP5 is not essential for *E. faecium* growth, it is essential for the resistance of *E. faecium* to β -lactam antibiotics (9). This characteristic of PBP5 led to the development of a disk diffusion susceptibility assay in which *E. faecium* growth depends on the expression of functional PBP5 (Fig. 3a). Briefly, all *E. faecium* PBPs except PBP5 are potentially inhibited by the β -lactam antibiotic ceftriaxone (PBP5 does not bind ceftriaxone and thus is not inhibited by it, Fig. S3a). Thus, a strain lacking the *pbp5* gene (*E. faecium* D344SRF, Figs. 3a-c; Fig. S3b) fails to grow in the presence of ceftriaxone, while *E. faecium* D344SRF transformed with a plasmid that expresses PBP5 is not sensitive to ceftriaxone enabling growth in the presence of the drug (Figs. 3a-c; Fig. S3b; PBP5-mediated resistance in *E. faecium* is enhanced when the gene encoding PBP5 is present downstream of genes *ftsW* and *psr*, both genes were incorporated into the plasmid upstream of *pbp5* (10)).'

This assays is based on the observation that *E. faecium* PBPs except PBP5 are inhibited by ceftriaxone (indeed, we agree with the reviewer that this sounds 'strange'; thus, we performed in vitro assays, summarized in Fig. S3a which show that ceftriaxone does not bind to PBP5, explaining while it fails to inhibit it). Thus, this assay allows us to identify mutants of PBP5 that render PBP5 unable to recruit and cross-link PG in amounts needed for *E. faecium* viability. When we add a vector containing the gene for wt-PBP5 to *E. faecium* D344SRF, it is fully viable as ceftriaxone does not bind PBP5. If the vector is empty - *E. faecium* D344SRF is not viable (its PBPs are inhibited by ceftriaxone). If we now add a vector containing the gene for *PBP5 variants* in which residues identified to be important for PG recruitment are mutated (based on our NMR interaction studies) and then test their ability to grow under the influence of ceftriaxone, we observe two distinct results. *E. faecium* is either viable (PG recruitment and, ultimately, crosslinking and cell wall formation function normally) or it is not viable (PG recruitment fails as the residues important for recruitment have been mutated). Because the variants selected were identified in our NMR experiments using PG/stem peptides as substrates (i.e. the CSPs report mechanistically on the recruitment) and reported on *recruitment* and not *cross-linking*, we are confident these results report on the recruitment. Finally, as an additional control, we confirm PBP5 wt and variant expression with a PBP5-specific antibody (Fig. 3c). Despite this, we have altered the language and the final sentence in this section as follows:

Together, these results strongly suggest that PBP5 residues Asn523, Asp530, Leu533, Trp457 and the residues of loop 625-631 are crucial for *E. faecium* PG/SP recruitment, and, further, that this is an obligatory step that precedes transpeptidation.

• **Figure 4**

o **For variant SP4, why was L-Lys(D-iAsn) replaced with Gly? Due to the lack of side chain resulting in extreme flexibility of Gly, this can result in unpredictable effects for functional interpretation. We instead suggest utilizing Ala in this position to preserve**

stereochemistry. The author may acetylate the amine of the iAsn moiety to isolate the role of the amine on the iAsn. Other strategies include using a peptide with different stereochemistry to clearly show the role of these residues.

We created 5 distinct SP variants to test different aspects of binding to PBP5. One was the complete replacement of the L-Lys(D-iAsn) moiety by substituting it with Gly, which lacks a sidechain. A Gly (vs Ala) residue was selected because our NMR STD experiments showed that the CH₂ moieties of L-Lys are contributing to the interaction. Thus, because Ala, and not Gly, contains a CH₃ moiety, we selected a Gly to create a fully binding deficient control. We agree with the reviewer that Gly likely has increased flexibility compared to Ala, but since these are very short peptides that have intrinsically very high dynamics, we felt this was a less significant concern. Lastly, we also created SP5 (L-Lys that lacks D-iAsn) to further test this role of the modified sidechain in binding.

Taken together, our extensive data provide convincing evidence of the molecular elements that mediate binding. We agree that if we had only done a single Gly variant it would not be sufficient, but since we complemented the Gly variant with 4 additional variants, our data rigorously describes the molecular basis of the interaction.

o A change of D-iGln to D-nGln resulted in a change in CSP highlighting the participation of this residue in the recruitment. Can the authors consider using a better control such as L-iGln or a more drastic change in the charge state of that amino acid by using D-isoglutamic acid or D-glutamic acid to strengthen that conclusion.

We agree with the reviewer that our PBP5 interaction with SP6 showed that this residue, as expected from the SP1 data, participates in the interaction (although less than the L-Lys sidechain). It might be of interest to create additional variants to further study this highly specific interaction, but we strongly feel this request is beyond the scope of the current work, where we focus on understanding the biological function of the PBP5 and PG interaction.

o The authors used SP2 to show that the terminal D-Ala-D-Ala is not necessary for recruitment. In general, the authors fail to clarify whether these peptides are recruited as acyl donor or acceptors. Does the pentapeptide and the tripeptide function as both donor and acceptor? Prior literature has shown that fluorescently labeled probes without the terminal D-ala-D-ala are able to label the PG. Are the authors challenging or substantiating these claims by NMR?

The reviewer is correct – our NMR data shows that there are no large changes for the interaction between PBP5 and either SP1 or SP2, directly showing that the D-Ala-D-Ala group is not critical for peptide binding. The reviewer highlights that ‘Prior literature has shown that fluorescently labeled probes without the terminal D-ala-D-ala are able to label the PG’ – indeed these results are completely consistent with our data that the D-Ala-D-Ala group is not important for binding to PBP5 (importantly, we are not testing the role of D-Ala-D-Ala for catalysis; clearly, D-Ala-D-Ala is important for catalysis). Thus, our data showing the lack of a role for D-Ala-D-Ala in PBP binding supports and is correlated with previous fluorescent labeled data.

Minor concerns

• We think the authors should improve the clarity of the manuscript by correcting grammar mistakes and improving the clarity of their conclusions to help the readers understand the story. For instance, “...suggesting that the mechanism underlying the efficacy of PBP inactivation by β -lactam antibiotics is not exclusively rely on substrate binding mimicry.” (pg. 11)

Thank you; we have updated the sentence as requested and changed 'is' to 'does'. We have also revised the manuscript to enhance readability and clarity.

• The authors refer to SP as “stem pentapeptides” while some of the peptides used here are not pentapeptides (SP2 and SP3). Please clarify these abbreviations

We thank the reviewer for pointing out this inconsistency. As requested, we changed SP to stem peptides to have consistency in the manuscript.

• Figure 2a

o We believe the authors should further discuss the necessity for the reduction of the isolated stem peptide in the text. If the reduced PG is used for technical/experimental purposes for the NMR, this needs to be explained.

The reduction of PG is not used for technical/experimental purposes for our NMR experiments. Testing both reduced and oxidized peptidoglycan fragments offered the possibility to test two mimics of the natural substrates instead of one.

o What is “bh” above the arrow in the figure? If referring to sodium borohydride, please use the chemical formula (NaBH₄).

Thank you - updated as requested.

• Figure 3

o We suggest placing labels on the gel rather than in the figure caption in order to increase the clarity

Thank you for highlighting this issue to us, but labels are already given in Figure 3d directly underneath the gel, as well as in the figure caption. We feel that naming the residues 3-times would be too busy and obscure the gel, i.e. the raw data.

• Figure 4

o What is “nGln” in variant SP6? Please include the structure of this

Thank you; “n” stands for “normal” and refers to “normal” amide bond found in proteins. We are not the ‘biggest fans’ of this nomenclature as well and have instead changed it to D-Gln.

o Why are variants SP4 and SP5 included in Figure 4 when it isn’t mentioned until Figure 5?

The reviewer is correct, SP4 and SP5 are not leveraged until Figure 5, but they are shown in Figure 4. This is done intentionally, as we felt that it is important to present all the peptides in a single figure to make it easier for readers to readily identify the differences between the SPs used in the study (vs, say, keeping only SP1-3 in Figure 4 and the SP4/5 in Figure 5, which forces the reader to go between figures to identify these differences). After comparing multiple drafted versions, we concluded that it is still more effective to show the difference between all SPs in one figure.

• Figure 5e

o The authors include a label for “L-Gly”. However, since Gly has no stereocenter, this should simply be “Gly”.

Thank you – updated as suggested.

• On pg. 12, the figure references for the following sentence are incorrect. It should refer to the supplemental figure S10, as there is no Fig. 10 in the main text: “Using 1D 1H STD NMR experiments we showed that SP6 has an interaction pattern similar to that of SP1 and includes protons from C β in D-iGln (Figs. S10f,g).

Thank you – updated as requested.

Reviewer #3 (Remarks to the Author):

I have a simple question: does the PBP5 preparation exhibit any enzymatic activity (transpeptidation and/or hydrolysis) on peptide SP1? If it does, it might be significant at the large PBP5 concentrations used.

As well described in the field – PBPs, such as PBP5, do not have any significant activity that can be readily observed in vitro. To ensure that this is also the case for the *E. faecium* PBP5 used in this study, we performed in vitro cross-linking experiments with purified components and tested the formation of peptidoglycan cross-links by high resolution mass spectrometry, as previously described (<https://doi.org/10.1074/jbc.M507384200>). As anticipated, no cross-linked products were identified. The reason why these enzymes are so inefficient in vitro is not completely understood, but it is completely consistent with literature reports for the last 20-25 years.

By the way, how long did the recordings of the NMR spectra take?

Thank you for this question, as it is indeed not easy to work with a protein of this size (and indeed very costly). Overall, about 70 weeks of 800 MHz NMR time are necessary for data collection, highlighting the technical difficulty but moreover the drive to high quality data (rigor) for such a large protein.

Reviewer #4 (Remarks to the Author):

Thank you for your efforts.

As a Journal of Biological Chemistry (JBC) Associate Editor, I would recommend looking into the specific program that we developed for early career reviewers at the JBC (<https://www.jbc.org/ecr>; <https://www.asbmb.org/asbmb-today/careers/100119/jbc-launches-program-for-early-career-scientists>) – I hope this ‘JBC-promotion plug’ is allowed here.

Reviewer #1 (Remarks to the Author):

I appreciate the detailed response to my comments, along with the additional data, clarifications, and manuscript revisions. Several of my comments were addressed, including inclusion of BMRB accession numbers and source data. Other important questions remain unanswered and are outlined below.

Thank you.

(1) The figures illustrating 1H-15N CSP plots do not appear to align with the source data. For instance, in Figure 1E, the annotated bars with residue identities do not show significant differences when compared to the source data. Moreover, the HSQC spectra, with chemical shifts from BMRB entry 51690, also reveal little to no changes for these residues. These discrepancies need to be explained.

We would like to emphasize that the CSP plots *were generated directly from the source data*, and there is no discrepancy. Figure 1E was plotted starting from residue 349 to focus on the TP domain, while the source data includes CSP values beginning from residue 35. This difference in residue numbering may have given the impression of a mismatch, but the plots are fully consistent with the underlying data.

We want to highlight that the BMRB data as well as the source data leverage the consistent numbering of the PBP5 residues and includes all data (i.e. not just the data for the TP domain).

One possibility for this discrepancy might be how the data is accessed/downloaded from the BMRB. When the data is downloaded as a simulated backbone HSQC (most convenient), the BMRB will report the numbering starting with residue 2 (independent on any comment(s) in the BMRB file). In general, we have seen that downloading the str file, which reports the correct residue numbering that is fully consistent with UniProt numbering and all figures in the manuscript, seems to be, at least in our hands, the easiest way to overcome this issue.

(2) For some residues (eg 523N) it is difficult to discern the spectral changes due to significant overlap with other signals. How were CSPs measured for overlapping peaks? Providing separate spectra for each titration point may illustrate this point.

We thank the reviewer for this comment. Because of the large size of the protein (~70 kDa), some regions of the spectra exhibit severe overlap, making it difficult to discern peak movements by direct visual inspection. To address this, we inspected the NMR spectra at each titration point, expanding crowded regions and carefully tracking peak positions. Importantly, CSP values were only calculated when peak shifts could be followed reliably across the titration.

Furthermore, for critical residues, such as N523, we also created single point variants (N523A) and expressed the protein in D₂O-based M9 medium, purified, refolded and re-recorded the 2D (¹H,¹⁵N) TROSY spectrum to ensure that N523 is correctly assigned – this is clearly the most through way of performing this analysis.

Lastly, the request to provide residue-by-residue spectra would amount to ~600 plots and is therefore impractical; we instead provide (i) the full overlay spectra for all titration points, which illustrate how peaks move globally, and (ii) CSP plots vs. residue, which summarize the quantified shifts. We believe this combination presents the data clearly while ensuring that CSP measurements are reliable even in overlapped regions.

(3) Are the data from the ILV 13C-1H spectra plotted correctly? They seem to be presented as a bar graph rather than as an XY plot, and the residue numbers are not continuous. In figure 1F, the numbers jump from 367 to 427 (an increase of 60), then to 449 (an increase of 22). This is confusing and the plots do not appear to accurately map CSP magnitudes to residue number. Also, the source data lists two values for the same residue in several cases. How are these depicted in the figure?

We thank the reviewer for her/his input and want to highlight that the 2D (1H,13C) ILV HSQC spectrum is shown in Supplementary Figure 1.

We checked that the ILV ¹H/¹³C CSP data are plotted correctly. Unlike backbone N/H^N plots, ILV methyl groups are not sequential in the primary sequence (i.e. not every amino acid in PBP5 is an ILV), so the residue numbering in Figure 1F cannot ever be continuous. This explains the apparent “jumps” in residue numbers (e.g., from 367 to 427, then to 449).

In addition, certain ILV residues contain two methyl groups (e.g., Leu, Val), which give rise to two distinct ¹H signals with the same ¹³C chemical shift. In such cases, both CSP values are reported in the source data and are plotted side by side in the figure, exactly as listed.

Thus, the bar plots directly represent the CSP magnitudes for each ILV methyl group with residue numbers on the X-axis and CSP values on the Y-axis, fully consistent with the source data.

(4) Based on BMRB entry 51690 and the source data, residues G98 and G268 undergo a substantial shift in response to SP, showing that some of the biggest CSP values in the dataset occur outside the 349-673 region displayed in the manuscript. These changes are potentially important and should be discussed, but they are not mentioned in the manuscript.

All CSPs observed are carefully and fully described. It is established that CSPs can result from two effects – direct binding or indirect conformational or dynamic changes. As described in detail in our previous publication (<https://doi.org/10.1038/s41467-023-39966-5>), some of the observed CSPs are not direct effects of binding; rather these CSPs are an indirect consequence of TP domain binding events. Binding triggers allosteric changes in the protein's dynamics or conformation, which are propagated through the interconnected structure of PBP5, particularly via loops L1 and L2, leading to the observed CSPs in the N1 and nPB domains.

(5) My previous question (#12) was not addressed – why does reduced PG (muramitol form) produce greater spectral changes than the oxidized form? Do additional CSPs reflect a structural property of the reduced sugar, a difference in stability/solubility, or another factor?

We thank the reviewer for this question. The larger CSPs observed for the reduced PG (muramitol form) most likely arise from chemical and structural differences between the reduced and oxidized forms. Reduction of the aldehyde group at the sugar terminus might yield to a more flexible muramitol moiety with altered hydrogen-bonding capacity, which may allow different modes of interaction with the protein and thereby produce additional CSPs.

Critically, we note that both reduced and oxidized PG samples were prepared under identical conditions, and no major differences in solubility or stability were observed during the NMR titrations. Indeed, the laboratory of Michel Arthur has tested stability for days at RT and did not

see any change. Therefore, the differences in CSP patterns are unlikely to reflect experimental artifacts, but rather genuine differences in how the reduced form engages with the protein, i.e. the reduced form binds stronger to PBP5 than the oxidized form.

(6) My previous comment (#15) was not addressed – suggestion to include a chart of peak intensities for each amino acid. to provide an additional layer of information regarding binding. I believe this would help illustrate the effects more clearly.

We appreciate the reviewers' concern regarding the best display of the data. As stated in our last response – it was not apparent to us to which data the reviewer referred to. Throughout the manuscript, we have tried to display our data as uniformly and carefully as possible. This is clearly more challenging for a 645 aa protein. Thank you for your support.

(7) Thank you for the response to my comment (#18). However, I am still unclear about the rationale for the specific design of the PBP5 variants used in the viability assay. My original question was more related to how the authors decided on the six particular constructs tested, especially given that some variants contain a single mutation while others include two or more. For example, N523A alone had no effect, whereas N523A/Q627A did. However, the Q627A single mutant was not tested. Please clarify how these specific combinations were chosen, and why certain potentially informative single mutants (such as Q627A) were not examined.

Thank you for your question. PBP5 variants selected correspond to the residues whose peaks exhibit significant changes in our NMR titration studies. The reviewer is correct, N523A did not provide a robust physiological response, so we added Q627A and created a double variant. The reviewer is correct that we did not test Q627A alone as it was clear that it was contributing to binding (explaining the effect of the N532A/Q627A variant). Next, we created a loop 625-631 variant, where all residues in loop 625 were mutated, which also includes residue Q627, which showed a strong physiological effect.

Lastly, and as shown in Figure 3d, residues were combined into a single variant readout when they are close in proximity in space. Taken together, these variants were selected based on the NMR interaction data.

Reviewer #2 (Remarks to the Author):

My concerns have been addressed.

Thank you for your support.

Reviewer #4 (Remarks to the Author):

Thank you for your help in reviewing.